# ATM phosphorylation of the actin-binding protein drebrin controls oxidation stress-resistance in mammalian neurons and *C. elegans*

Patricia Kreis[1], Christian Gallrein[2], Eugenia Rojas-Puente[1], Till G.A. Mack[1], Cristina Kroon[1], Viktor Dinkel[1], Claudia Willmes[1], Kai Murk[1], Susanne tom-Dieck[3], Erin M. Schuman [3], Janine Kirstein [2] & Britta J. Eickholt[1,4]

Drebrin (DBN) regulates cytoskeletal functions during neuronal development, and is thought to contribute to structural and functional synaptic changes associated with aging and Alzheimer's disease. Here we show that DBN coordinates stress signalling with cytoskeletal dynamics, via a mechanism involving kinase ataxia-telangiectasia mutated (ATM). An excess of reactive oxygen species (ROS) stimulates ATM-dependent phosphorylation of DBN at serine-647, which enhances protein stability and accounts for improved stress resilience in dendritic spines. We generated a humanized DBN *Caenorhabditis elegans* model and show that a phospho-DBN mutant disrupts the protective ATM effect on lifespan under sustained oxidative stress. Our data indicate a master regulatory function of ATM-DBN in integrating cytosolic stress-induced signalling with the dynamics of actin remodelling to provide protection from synapse dysfunction and ROS-triggered reduced lifespan. They further suggest that DBN protein abundance governs actin filament stability to contribute to the consequences of oxidative stress in physiological and pathological conditions.

[1] Institute of Biochemistry, Charité – Universitätsmedizin Berlin, Charitéplatz 1, 10117 Berlin, Germany. [2] Leibniz-Research Institute for Molecular Pharmacology (FMP), Robert-Roessle-Straße 10, 13125 Berlin, Germany. [3] Max Planck Institute for Brain Research, Max-von-Laue Strasse 4, 60438 Frankfurt, Germany. [4] NeuroCure – Cluster of Excellence, Charité – Universitätsmedizin, 10117 Berlin, Germany. These authors contributed equally: Patricia Kreis, Christian Gallrein, Eugenia Rojas-Puente. Correspondence and requests for materials should be addressed to P.K. (email: patricia.kreis@charite.de) or to J.K. (email: kirstein@fmp-berlin.de) or to B.J.E. (email: britta.eickholt@charite.de)

Drebrin (developmentally regulated brain protein, DBN) is a conserved F-actin side-binding protein that reduces actin filament turnover[1–3]. It is particularly enriched in dendritic spines, where it is thought to control spine morphology and function[4]. Since progressive loss of DBN in the brain has been correlated with cognitive deficits associated with ageing, DBN has been postulated to mediate protection against ageing-induced dendritic spine degeneration[5–9]. Paradoxically, normal dendritic spine shape and synapse function has been reported in *Dbn*−/− brains, suggesting that DBN-loss alone is not sufficient to induce synapse dysfunction[10]. We thus set out to explore here if cellular insult, stress, or ageing renders neurons vulnerable to DBN-loss.

## Results

**Increased vulnerability of dendritic spines in *Dbn*−/− neurons.** We treated hippocampal neurons of wild-type (WT) and *Dbn*−/− mice with different stressors (Fig. 1) and analysed dendritic spine density using a stringent, semi-automated analysis (Fig. 1a). Initially, we induced depolymerization stress in actin filaments using latrunculin (LatB). At low doses (1 μM), LatB did not affect dendritic spine density in control neurons, whilst at higher concentrations (5 μM), it induced a significant reduction in spine density (Fig. 1b). We then examined the effects of low LatB concentrations on spine density in neurons isolated from DBN-deficient mice[10], showing that compared to WT vehicle controls, neurons with decreased levels of DBN (Fig. 1c) responded to 1 μM LatB with significant decreases in spine density (Fig. 1d). This result indicates that reducing the level of DBN affects F-actin content in dendritic spines, a process that is unmasked by low concentrations of LatB.

Excessive dendritic spine loss is one of the earliest events in Alzheimer's disease (AD), and is triggered by $A\beta_{1-42}$ oligomers[11]. To analyse if loss of DBN increases susceptibility to the synaptotoxic effects of $A\beta_{1-42}$, we challenged hippocampal neurons with amyloid peptide oligomeric preparations[12]. At concentrations that had previously been demonstrated to induce spine loss in rat hippocampal neurons (1 μM or lower)[13,14], the amyloid peptide did not induce significant decreases in spine density in our mouse hippocampal neurons. We believe this discrepancy is likely due to species differences (rat vs. mouse), or normal variations in neuron preparations or peptide oligomerisation. However, we found a significant reduction in spine density in *Dbn*−/− neurons with 1 μM amyloid peptide (Fig. 1e), indicating that drebrin loss renders spines more vulnerable towards synaptotoxic effects of $A\beta_{1-42}$. $A\beta_{1-42}$ has been demonstrated to evoke chronic toxic effects associated with AD by increasing reactive oxygen species (ROS) production[15]. To test if chronic oxidative stress alone is sufficient for unmasking the system's vulnerability, we applied paraquat (*N*,*N*′-dimethyl-4,4′-bipyridinium dichloride; PQ) to chronically generate superoxide radicals[16]. In WT neurons, exposure to PQ induced a moderate reduction in spines, whilst *Dbn*−/− neurons responded to PQ with a substantial decrease in spine density (Fig. 1f). Taken together, these experiments demonstrate that reductions in DBN protein levels increase susceptibility to AD and oxidative stress. In a final step in this series, we determined whether oxidation-induced defects in spine density of *Dbn*−/− neurons could be rescued by re-expressing DBN. DBN re-expression using lentiviral infection induced a slight increase in spine density in control conditions and further protected synapses from both PQ-induced spine loss, and PQ-induced reduction in F-actin content in spines (Fig. 1g). These results suggest that levels of DBN protein may orchestrate actin filament stability to prevent loss of synapses in response to oxidative stress.

**DBN stability is regulated by phosphorylation and ubiquitination.** To identify the biochemical processes responsible for regulating cellular DBN protein levels, we considered the role of phosphorylation in the control of DBN protein turnover. Examination of DBN expression and phosphorylation in tissue homogenate obtained from rat forebrain revealed strikingly similar developmental patterns of pS647-DBN when compared to the pan-DBN (Fig. 2a). In contrast, pS142-DBN, a phosphorylation event induced by CDK-5[17], decreased sharply during early developmental stages and was absent from the adult (higher molecular weight) DBN A isoform, indicating independent control of DBN phosphorylation at S647 and S142 during brain development. We also observed strong co-localization of pS647-DBN and total DBN in mouse hippocampus by immunohistochemistry. Direct measurements of fluorescence intensities revealed strong correlation between DBN puncta and their corresponding pS647-DBN signals (Fig. 2b). Collectively, these results indicate that pS647-DBN directly correlates with relative DBN protein abundance, prompting us to analyse if phosphorylation at S647 contributes to the regulation of DBN protein stability.

To study DBN degradation kinetics in terms of dependence of pS647, we exploited the non-natural amino acid azidohomoalanine (AHA), which has been shown to effectively incorporate into newly synthesized proteins in place of the amino acid methionine[18]. AHA-labelled proteins can be covalently coupled to an alkyne-bearing affinity tag in a click-chemistry reaction and identified by biochemistry or by histochemistry[19]. We transfected human embryonic kidney (HEK) 293T cells with WT FLAG-DBN or FLAG-DBN mutants mimicking phosphorylation (S647D) or de-phosphorylation (S647A), and pulse-labelled methionine-starved cells with AHA for 1 h. Pulse labelling of proteins was followed by a chase with unlabelled amino acids and the proteins were subsequently detected by chemo-selective ligation and streptavidin labelling. $DBN^{S647A}$ mutant exhibited decreased protein stability when compared to $DBN^{S647D}$ and DBN WT protein (Fig. 2c, d). Calculation of protein half-life indicated that DBN WT is long-lived with a half-life of approximately 3 days ($75 \pm 8$ h), whereas $DBN^{S647A}$ increased protein turnover by approximately 50% ($39 \pm 3$ h). Given that the protein half-life of DBN does not differ significantly from the half-life of $DBN^{S647D}$ ($75 \pm 8$ h vs. $97 \pm 16$ h), our experiments suggest that phosphorylation at S647 protects DBN from degradation and that DBN is highly phosphorylated on this site in cells. To test if DBN is degraded by the 26S proteasome, we performed the AHA pulse-chase assay in the presence of the proteasome inhibitor MG132, which showed that $DBN^{S647A}$ was stabilized, suggesting degradation by the ubiquitin–proteasome system (Fig. 2e). To directly examine modification of DBN by ubiquitination, HEK 293T cells were transiently transfected with FLAG-tagged DBN in combination with a hemagglutinin-tagged ubiquitin-green fluorescent protein (HA-Ub-GFP) construct. As shown in Fig. 2f, significant amounts of incorporated HA–protein conjugates were detected within the FLAG-DBN precipitates. Importantly, the levels of HA-incorporated DBN increased in the presence of MG132, indicating that proteasome-mediated degradation contributes to the control of DBN protein levels.

**Visualization of endogenous DBN turnover in neurons.** AHA is an effective tool to study newly synthesized proteins and, by combining this technique with click-chemistry, labelled, newly synthesized proteins can be tagged and visualized in situ using fluorescence (fluorescent non-canonical amino acid tagging, or FUNCAT)[20]. FUNCAT has recently been successfully combined

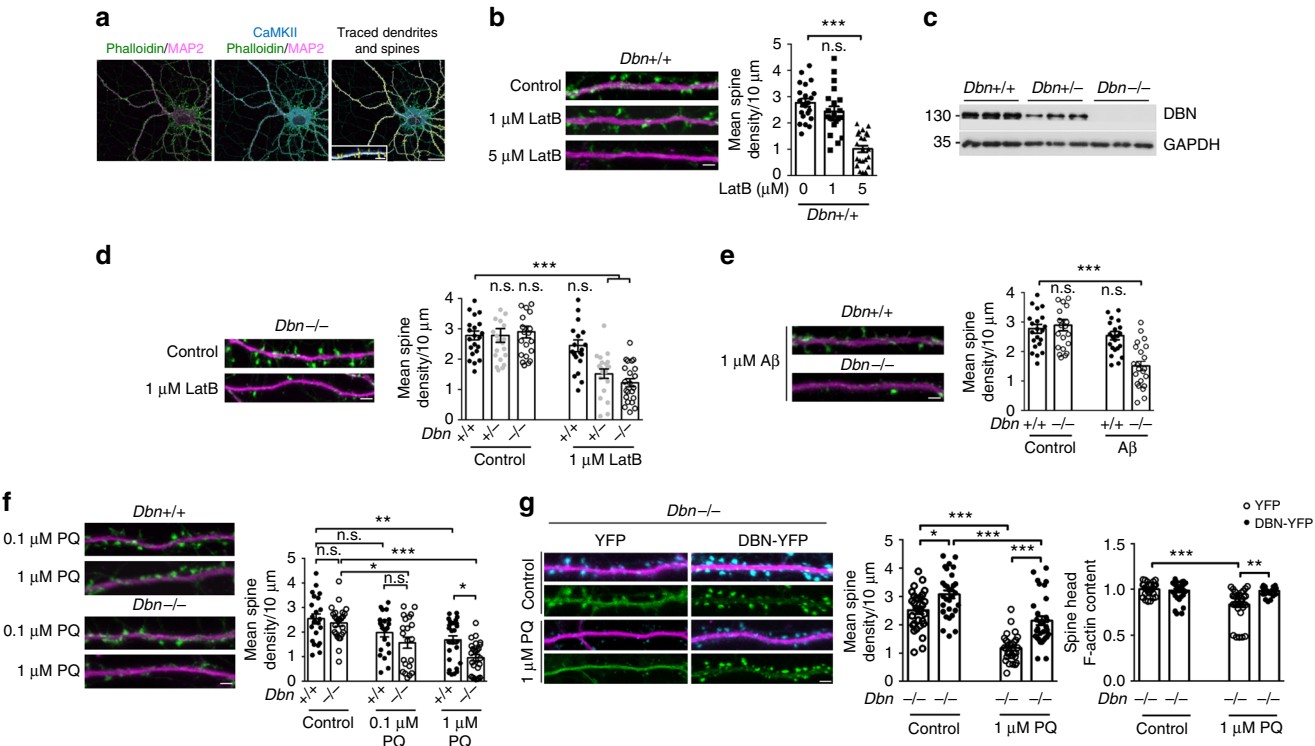

**Fig. 1** Increased vulnerability of dendritic spines in DBN-depleted neurons. **a** Spine density analysis using an automated detection method. Left: WT neurons labelled for dendrites (MAP2, magenta) and dendritic spines (Phalloidin, green). Middle: spines labelled with F-actin (green) and CaMKII (blue) and MAP2 (magenta). Right: detection of the entire dendritic tree (white) and the dendritic spines (yellow) by the Imaris software using MAP2 and Phalloidin labelling. Sub-threshold signals from the underlying dendritic network were ignored (insert). Scale bar: 5 μm; scale bar in insert: 3 μm. **b** Spine density analysis of WT neurons (*Dbn+/+*) treated with latrunculin B (LatB) for 2 h. Left: representative image of a dendrite. Scale bar: 3 μm. Right: mean spine density/10 μm ± SEM. N = 3. LatB 0 μM: n = 22, LatB 1 μM: n = 21, LatB 5 μM: n = 22. **c** DBN protein levels in adult mouse brain lysates of wild-type (+/+), heterozygote (+/−) or knockout (−/−) *Dbn* brains. Molecular weight protein ladder is in kilodaltons. **d** Spine density analysis of *Dbn+/+*, *Dbn+/−* or *Dbn−/−* neurons in the absence (control) or presence of 1 μM LatB. Left: representative image of a *Dbn−/−* dendrite. Right: mean spine density/10 μm ± SEM. N = 3. Control, *Dbn+/+*: n = 22, *Dbn+/−*: n = 21, *Dbn−/−*: n = 22. One micromole of LatB *Dbn+/+*: n = 21, *Dbn+/−*: n = 21, *Dbn−/−*: n = 22. **e** Spine density analysis of *Dbn+/+* and *Dbn−/−* neurons in the presence of 1 μM of oligomeric Aβ$_{1-42}$ for 24 h. Left: representative image of a *Dbn−/−* and a *Dbn+/+* dendrite in the presence of Aβ$_{1-42}$. Right: mean spine density/10 μm ± SEM. N = 3. Control, *Dbn+/+*: n = 22, *Dbn−/−*: n = 21. Aβ$_{1-42}$, *Dbn+/+*: n = 22, *Dbn−/−*: n = 22. **f** Spine density analysis of *Dbn+/+* and *Dbn−/−* neurons treated 24 h with paraquat (PQ). Left: representative image of a *Dbn+/+* and a *Dbn−/−* dendrite in the presence of paraquat. Right: mean spine density/10 μm ± SEM. N = 3. Control, *Dbn+/+*: n = 35, *Dbn−/−*: n = 31. 0,1 μM PQ, *Dbn+/+*: n = 35, *Dbn−/−*: n = 32. One micromole of PQ, *Dbn+/+*: n = 32, *Dbn−/−*: n = 31. **g** Spine density analysis of *Dbn−/−* neurons infected with YFP or DBN-YFP in the presence of PQ. Left: representative images of a *Dbn−/−* dendrite expressing YFP constructs (green), labelled with MAP2 (magenta) and Phalloidin (F-actin, cyan). Scale bar 3 μm. Middle: mean spine density/10 μm ± SEM. Right: spine head F-actin content relative to *Dbn−/−* control ± SEM. N = 3. Control, YFP: n = 30, DBN-YFP: n = 30. One micromole of PQ, YFP: n = 30, DBN-YFP: n = 31. Statistics were performed using one-way ANOVA Bonferroni test. *$p < 0.05$, **$p < 0.001$, ***$p < 0.001$, n.s.: non-significant. N = number of biologically independent experiments, n = total number of neurons. DBN developmentally regulated brain protein, ANOVA analysis of variance

with proximity ligation assay (PLA), generating the opportunity to visualize an endogenous protein of interest as it is newly synthesized in situ (FUNCAT-PLA)[21]. Here, we used FUNCAT-PLA to visualize endogenous DBN protein synthesis in rat hippocampal neurons, and to monitor DBN degradation in situ. Methionine-starved hippocampal neurons were AHA pulse-labelled and fixed. After click-chemistry reaction, numerous PLA signals were detected in the MAP2-positive dendrite compartment and in close proximity to dendrites, most likely reflecting dendritic spines (Fig. 3a, b). The presence of newly synthesized DBN PLA puncta in dendritic spines suggests that DBN is locally translated in these compartments, although local transport into the spine during labelling cannot be excluded. To confirm the specificity of the obtained signals, we applied the protein synthesis inhibitor anisomycin during AHA labelling or used methionine (instead of AHA) for pulse labelling, before initiating the FUNCAT-PLA detection protocol. In both cases, signals were barely detectable, supporting that obtained DBN

FUNCAT-PLA signals delineate newly synthesized protein (Fig. 3b). We then exploited FUNCAT-PLA to visualize endogenous turnover of DBN protein in situ and 'chased' cultures with AHA-free medium for different times (Fig. 3a). DBN FUNCAT-PLA signals declined over the examined time period, with 50% of the initial signal disappearing within 68 h (Fig. 3c). Thus, DBN protein stability in situ can be assessed with FUNCAT-PLA, and the data are consistent with half-life values determined by our biochemical analyses (Fig. 2d).

**DBN is phosphorylated by ATM at serine-647.** We have previously demonstrated an increase in phosphorylation of the actin-binding protein DBN at S647 by a yet uncharacterized kinase activity, in response to membrane depolarization following exposure to elevated levels of extracellular potassium[22]. Examination of the amino acid sequence revealed DBN-S647 as a putative consensus phosphorylation motif of the serine/threonine kinase ataxia-telangiectasia-mutated kinase (ATM)[23] (Fig. 4a).

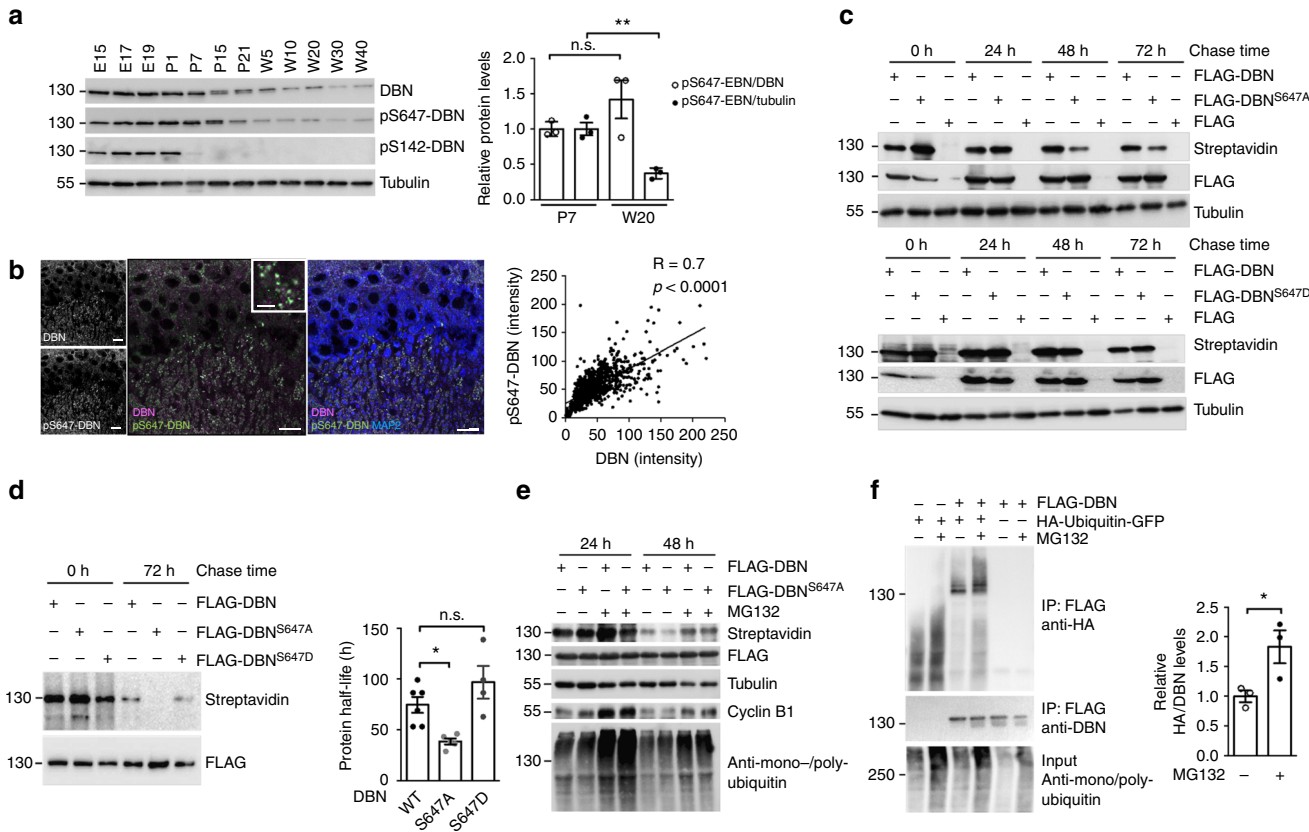

**Fig. 2** DBN stability is regulated by phosphorylation and ubiquitination. **a** Developmental expression of DBN and phospho-DBN in the rat brain. Left: western blot analysis of rat brain protein lysates. E: embryonic day; P: postnatal day; W: week. Molecular weight protein ladder is in kilodaltons. Right: quantitative analysis of pS647-DBN protein levels at P7 and W20 relative to DBN or to tubulin. $n = 3$ independent animals. Mean ±SEM. One-way ANOVA Bonferroni test. **$p < 0.01$. n.s.: non-significant. **b** Immunohistological analysis of DBN (magenta), pS647-DBN (green) and MAP2 (blue) at P7 in the CA3 region of the mouse hippocampus. Scale bars: 20 μm. Insert demonstrates the overlap of DBN and pS647-DBN immunolabelling in synapses. Scale bar insert: 10 μm. Right: correlative curve between the intensity of DBN and the intensity of pS647-DBN. $N = 4$, $n = 1044$. Spearman's correlation coefficients $r$ and $p$ values are indicated. **c** Analysis of DBN protein stability in terms of dependence of phosphorylation at S647. HEK 293T cells were transfected with FLAG constructs, labelled with AHA for 1 h and chased for 24, 48 or 72 h. Labelled protein was detected with streptavidin, total transfected protein with anti-FLAG antibodies (S647A top; S647D bottom) and total protein amount with anti-α-tubulin antibodies. **d** Determination of DBN wild-type, DBN^S647A and DBN^S647D protein half-life in HEK 293T cells. Mean ± SEM, *$p < 0.05$. WT: $n = 6$, S647A: $n = 5$, S647D: $n = 6$. $n =$ biologically independent experiments. One-way ANOVA Bonferroni test. *$p < 0,05$, n.s.: non-significant. **e** DBN degradation is proteasome-dependent. Pulse-chase assay in the presence of vehicle DMSO (−) or MG132 (+). Note the increase in cyclin B1 levels and in mono polyubiquitinated proteins in the presence of MG132. **f** DBN is ubiquitinated. HEK 293T cells expressing FLAG-DBN and HA-ubiquitin-GFP constructs were treated with DMSO (−) or MG132 (+). Left: western blot analysis of immunoprecipitated FLAG-DBN detecting HA-ubiquitinated conjugates. Right: quantitative analysis of HA conjugates relative to immunoprecipitated FLAG-DBN. $n = 3$ biologically independent experiments. Mean ± SEM. T test *$p < 0.05$. DBN developmentally regulated brain protein, DMSO dimethyl sulfoxide, ANOVA analysis of variance

Human mutations in the ATM gene are linked to ataxia-telangiectasia (A-T), a rare progressive neurodegenerative, autosomal recessive disease causing severe disability[24]. To determine whether ATM is able to phosphorylate DBN at S647 in vitro, HEK 293T cells were transfected with complementary DNA (cDNA) encoding FLAG-tagged ATM and treated with hydrogen peroxide ($H_2O_2$). Immunoprecipitated ATM was then incubated with recombinant His-chitin-tagged DBN fusion protein in the absence or presence of the specific ATM inhibitor KU55933[25]. Western blotting using a phospho-specific antibody recognizing pS647-DBN demonstrated phosphorylation of DBN by ATM, but not in the presence of KU55933 (Fig. 4b). Whilst ATM plays important functions in mediating cellular responses induced by DNA double-strand breaks, recent studies identified that cytosolic ATM pools also regulate essential redox signalling induced by oxidative stress[26]. To determine if oxidation-induced activation of ATM controls S647-DBN phosphorylation in cells, we used HEK 293T cells that express ATM and DBN endogenously. $H_2O_2$

induced a robust activation of ATM, as determined by phosphorylation of ATM substrate p53 (Fig. 4c). In parallel, we detected an increase in pS647-DBN, which was sensitive to KU55933. Since DBN is absent from the nucleus in HEK 293T cells, this is in agreement with an oxidation-induced activation of ATM in the cytoplasm that phosphorylates the actin-binding protein. Finally, to ascertain if ATM is amongst the brain-derived kinases that are able to phosphorylate DBN at S647, brain lysates were incubated with purified DBN protein. Under these conditions, but not when KU55933 was added to brain extracts, robust DBN-S647 phosphorylation was detected (Fig. 4d). These results demonstrate that activated ATM is present in postnatal rat brain, and that it can phosphorylate DBN.

**Neuronal activity increases oxidation in proximity of DBN.** Neurons are highly metabolically active cells that produce robust levels of ROS during neuronal activity[27]. In the healthy brain, ROS generation is effectively buffered by different antioxidant

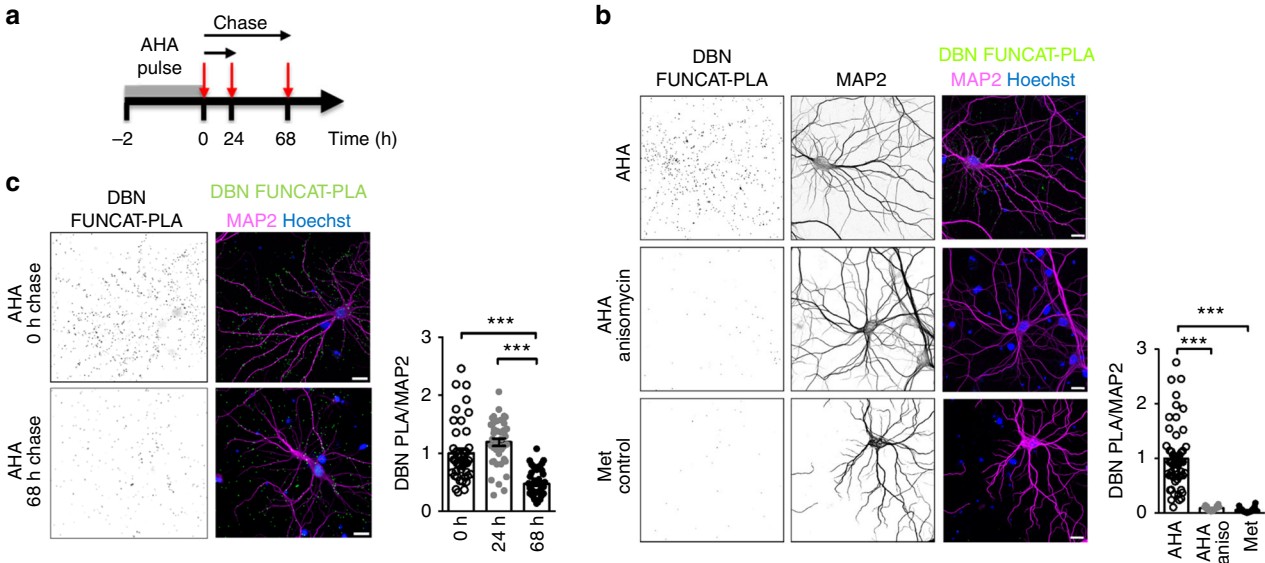

**Fig. 3** Visualization of endogenous DBN turnover in neurons using FUNCAT-PLA. **a** Schematic experimental protocol. Hippocampal neurons were incubated with AHA, and chased with AHA-free medium before performing FUNCAT click chemistry. Immunostaining of DBN and subsequent proximity ligation assay (PLA) performed in situ allow visualization of newly synthesized DBN. **b** Verification of the FUNCAT-PLA experimental protocol. Fluorescent newly synthesized DBN (FUNCAT-PLA in magenta) in the presence of AHA alone (AHA), AHA and anisomycin (AHA-anisomycin) or methionine alone (Met control). Quantification of the ratio of the FUNCAT-PLA signal over the cell volume marker area (MAP2), relative to AHA-treated neurons is as shown. AHA: $n = 53$, $N = 5$; AHA-anisomycin: $n = 14$, $N = 3$; Met control: $n = 32$, $N = 5$. **c** Newly synthesized DBN after 2 h pulse with AHA (0 h chase) followed by 24 h chase or 68 h chase. Quantification of the ratio of FUNCAT-PLA signal over cell volume marker area (MAP2), relative to the time-point 'chase 0 h' is as shown. 0 h chase: $n = 42$, $N = 3$; 24 h chase: $n = 40$, $N = 3$; 68 h chase: $n = 44$, $N = 3$. All bar graphs are represented as mean ± SEM. Statistics were performed using one-way ANOVA Bonferroni test. ***$p < 0.001$. $N =$ number of biologically independent experiments, $n =$ total number of neurons. DBN developmentally regulated brain protein, ANOVA analysis of variance

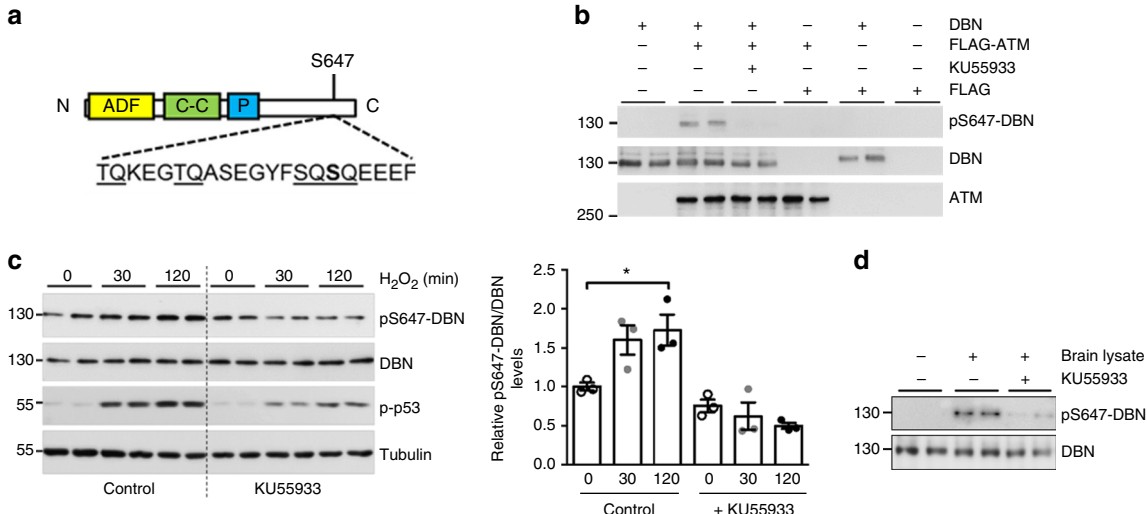

**Fig. 4** DBN is phosphorylated by ATM at S647. **a** DBN domain structure consists of an N-terminal ADF/cofilin homology domain (ADF), a coiled-coil (C–C) domain and a proline-rich stretch (P). The amino acid sequence surrounding S647 is characterized by several SQ and TQ motifs. **b** ATM in vitro kinase assay. Purified FLAG-ATM or FLAG were incubated with recombinant DBN in the absence (−) or presence (+) of the ATM inhibitor KU55933. Molecular weight protein ladder is in kilodaltons. **c** HEK 293T cells were exposed to $H_2O_2$ (250 µM) in the presence of DMSO (control) or 10 µM KU55933. Left: representative western blot analysis of pS647-DBN, DBN and tubulin. Levels of phosphorylation of p53 were used as control for ATM activity. Right: Bar graph shows the relative band density of pS647-DBN/DBN in three biologically independent experiments ± SEM; One-way ANOVA Bonferroni test. *$p < 0.05$. **d** In vitro kinase assay using recombinant DBN and adult mouse brain extract in the presence of DMSO (−) or KU55933 (+). DBN developmentally regulated brain protein, ATM ataxia-telangiectasia mutated, DMSO dimethyl sulphoxide, ANOVA analysis of variance

systems. However, under specific conditions, such as ageing, hyperactivity or toxic insults, ROS production can be detrimental and induces oxidation and damage of essential macromolecules such as enzymes and structural proteins[28]. We fused reduction-

oxidation sensitive GFP (roGFP) to DBN (DBN-roGFP) to monitor the redox conditions in the immediate proximity of DBN. This redox sensor displays dual excitation at 405 and 488 nm, with a single emission at 509 nm[29]. Oxidizing conditions

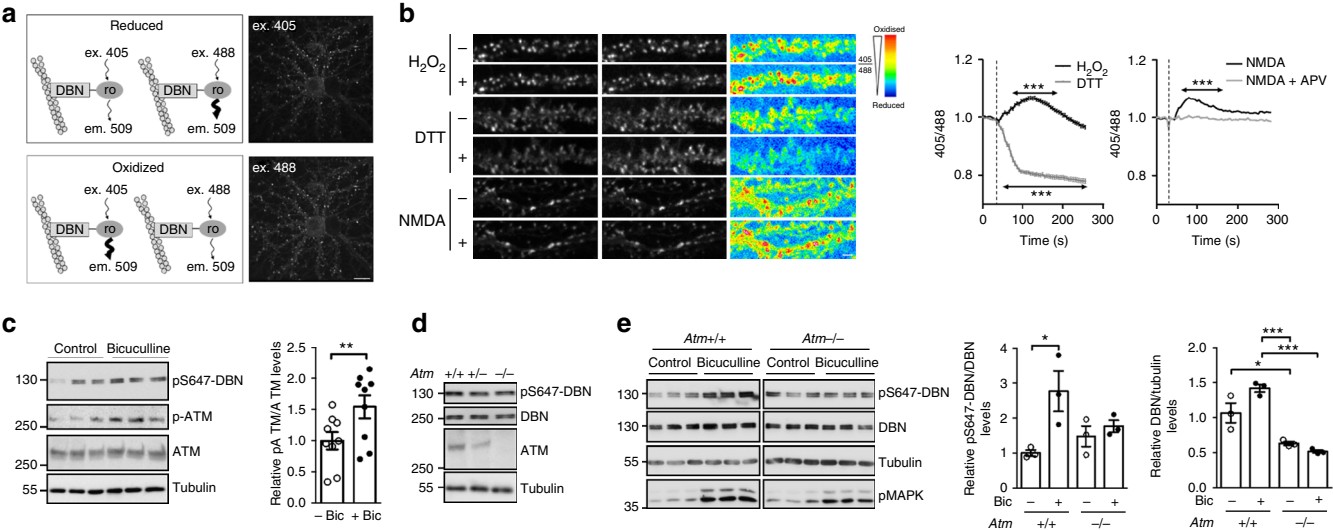

**Fig. 5** ATM-dependent phosphorylation of DBN at S647 is controlled by neuronal activity. **a** Analysis of the redox environment in dendritic spines using DBN-roGFP. Left: excitation/emission properties of DBN-roGFP. roGFP has two excitation peaks at ~405 and ~488 nm and a single emission peak at 509 nm. Reduced conditions result in a higher excitation at ~488 nm and a lower excitation at ~405 nm. The reverse occurs under oxidized conditions. Right: DIV21 hippocampal neurons expressing DBN-roGFP excited at 405 nm (upper neuron) or 488 nm (lower neuron) and detected at 509 nm. Scale bar 15 μm. **b** Neuronal activity changes the redox environment in the proximity of DBN. Left: Representative dendrites of DIV21 hippocampal neurons expressing DBN-roGFP excited at 405 nm (first column), 488 nm (second column) or false-coloured 405/488 ratios (third column) before (−) and after (+) treatment. Scale bar 3 μm. Right: mean 405/488 ratio traces ± SEM. $H_2O_2$, DTT or NMDA were applied as indicated by a dotted vertical line. $H_2O_2$: $n = $ 125 spines from 13 neurons of three independent experiments; DTT: $n = 71$ spines from 10 neurons of three biologically independent experiments; NMDA: $n = 94$ spines from 11 neurons of three biologically independent experiments; NMDA + APV: $n = 88$ spines from 11 neurons of three biologically independent experiments. One-way ANOVA Bonferroni test. ***$p < 0.001$ compared to $T = 0$. **c** Hippocampal neurons DIV14 were treated with 30 μM bicuculline for 1 h before lysis. Bar graph shows the relative band density of pATM/ATM in nine independent experiments ± SEM; $T$ test. **$p < 0.01$. **d** Western blot analysis of $Atm+/+$, $Atm+/−$ or $Atm−/−$ hippocampal neuron (DIV 20) lysates. **e** Hippocampal neurons (DIV14) were treated with 30 μM bicuculline for 1 h before cell lysis. The increase in pMAPK was used as control for bicuculline-induced network excitation. Bar graph shows the relative band density of pS647-DBN/DBN (left) and DBN/tubulin (right) in three biologically independent samples ± SEM; One-way ANOVA Bonferroni test. *$p < 0.05$; ***$p < 0.001$. ATM ataxia-telangiectasia mutated, DBN developmentally regulated brain protein, ANOVA analysis of variance

result in higher excitation at 405 nm and lower excitation at 488 nm, whereas the reverse occurs in reducing conditions (Fig. 5a). Initially, to ensure that DBN-roGFP is able to report the redox status within a dynamic range, we performed live cell imaging in COS cells transfected with DBN-roGFP exposed either to $H_2O_2$ or dithiothreitol (DTT), and by exciting the cells at 405 and 488 nm, we measured the emission at 509 nm. Following exposure to $H_2O_2$ or DTT, cells responded with an increase or a decrease in the ratio 405/488, respectively (Supplementary Figure 1). These data demonstrate DBN-roGFP as a suitable sensor to assess changes in ROS levels. We then infected hippocampal neurons with virus, which produced bright signals of DBN-roGFP in dendritic spines (Fig. 5a). Exposure of neurons to $H_2O_2$ led to a brief, significant increase of the 405/488 ratio locally in dendritic spines (Fig. 5b). Conversely, neurons exposed to the reducing agent DTT shifted the ratio to lower 405/488 values. Finally, we tested if increasing neuronal activity can modify the redox status of DBN-roGFP in neurons. Exposing neurons to 50 μM of N-methyl-D-aspartic acid (NMDA) led to a small, yet significant increase of the 405/488 ratio, which could be antagonized using the NMDA receptor antagonist amino-5-phosphonovaleric acid (APV) (Fig. 5b). These results demonstrate that under conditions of increased neuronal activity, the immediate DBN environment experiences moderate oxidation.

**$DBN^{S647D}$, but not $DBN^{S647A}$, rescues the loss of DBN or ATM.** We next tested if increased activity is able to promote ATM activation in neurons, and treated cortical neurons with the γ-aminobutyric acid (GABA) receptor antagonist bicuculline, which is known to increase neuronal firing and synaptic activity,

as well as to increase ATM substrate phosphorylation (at the so-called SQ/TQ motifs)[30]. Western blotting of neuronal cell lysates identified increased S1981 phosphorylation levels of ATM, an indicator of ROS-induced activation of the kinase[26] (Fig. 5c). We then asked if neuronal activity-induced activation of ATM results in increased phosphorylation of DBN. Western blotting of neuronal cell lysates revealed increased DBN-S647 phosphorylation following exposure to bicuculline, which was effectively blocked by KU55933 (Supplementary Figure 2). To substantiate these findings, we used mice deficient in ATM[31]. $Atm−/−$ mice and their WT littermates express equivalent levels of DBN protein and demonstrate comparable levels in pS647-DBN in the brain, suggesting that ATM does not control steady-state DBN phosphorylation (Fig. 5b). In contrast, activity-induced S647 phosphorylation of DBN by using a GABA type A (GABA-A) receptor antagonist bicuculline is blocked in $Atm−/−$ neurons (Fig. 5e). These findings indicate that DBN may be a convergence point for ATM functions in neurons and implicates this kinase in the control of DBN function in response to ROS production. Given that DBN resides exclusively in the post-synaptic compartment[4], this mechanism further points towards a function of ATM in dendritic spines that may function to provide a safeguard buffer to avert synapse shrinkage during challenging conditions. In this case, ATM-loss should phenocopy $Dbn−/−$ and sensitize neurons during stress-induced spine loss. To test this idea, we treated hippocampal neurons of WT and $Atm−/−$ mice with PQ, as previously described (Fig. 1f, g) and analysed dendritic spine density. In the absence of ATM, oxidation-induced reduction in spine density was exacerbated (Fig. 6a). Next, we examined the effects of DBN phosphorylation on oxidation-induced loss of

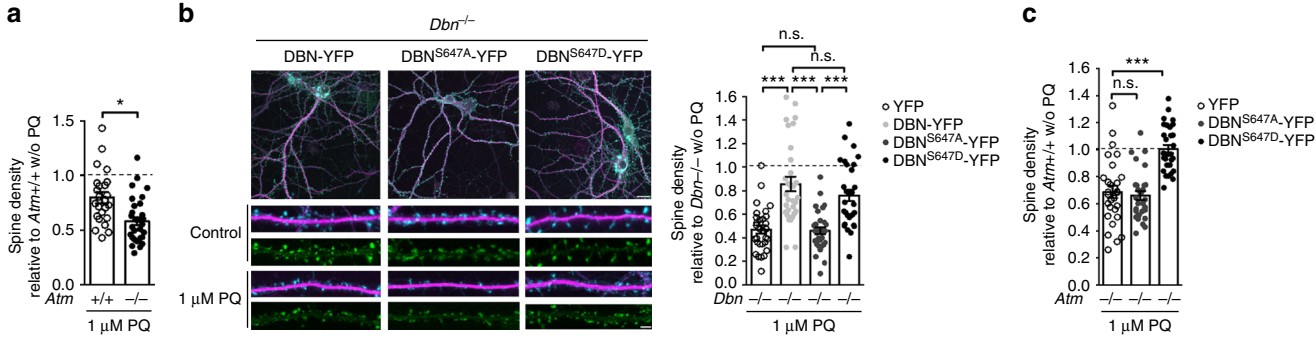

**Fig. 6** DBN^S647D, but not DBN^S647A, rescues the loss of DBN or ATM. **a** Spine density analysis of $Atm+/+$ or $Atm-/-$ hippocampal neurons treated with PQ for 24 h. ATM+/+: $N = 3$, $n = 24$; ATM $-/-$: $N = 3$, $n = 31$. **b** Spine density analysis of $Dbn-/-$ hippocampal neurons infected with YFP constructs in the presence of PQ. Left: representative overlay of a neuron expressing DBN-YFP (green) labelled with MAP2 (magenta) and Phalloidin (F-actin, cyan). Upper images: scale bar 20 μm. Lower images: scale bar 3 μm. Right: bar graph presents spine density relative to DBN knockout controls. $N = 3$. YFP: $n = 30$, DBN-YFP: $n = 31$, DBN^S647A-YFP: $n = 32$, DBN^S647D-YFP: $n = 30$. **c** Spine density analysis of ATM $-/-$ hippocampal neurons infected with YFP constructs in the presence of PQ. $N = 3$. YFP: $n = 29$, DBN^S647A-YFP: $n = 30$, DBN^S647D-YFP: $n = 31$. All statistics were performed using one-way ANOVA Bonferroni test. *$p < 0.05$, ***$p < 0.001$. n.s.: non-significant. $N =$ number of biologically independent experiments, $n =$ total number of neurons. DBN developmentally regulated brain protein, ATM ataxia-telangiectasia mutated, ANOVA analysis of variance

spines and performed rescue of $Dbn-/-$ using DBN WT, DBN^S647A and DBN^S647D. The results indicate that, whilst expression of DBN WT and DBN^S647D were both able to compensate for the loss of endogenous DBN, expression of DBN^S647A did not modify the induced loss of spines suggesting that phosphorylation at S647 may be a key feature of the protective effect of DBN to sustain spine integrity during stress (Fig. 6b). Finally, we examined the effect of overexpressing DBN phosphorylation mutants in $Atm-/-$ neurons during oxidative conditions. In hippocampal neurons of $Atm-/-$ mice treated with PQ, dendritic spine loss was rescued to control levels following overexpression of DBN^S647D, but not DBN^S647A. (Fig. 6c). These results demonstrate that ATM contributes to withstanding synapse shrinkage during challenging conditions by controlling DBN phosphorylation.

**Serine-647 DBN regulates lifespan and stress resistance in Caenorhabditis elegans.** To determine if S647 phosphorylation underlies ATM-dependent protection from oxidative stress in vivo, we employed the ageing model organism Caenorhabditis elegans. We first tested the ΔDBN-1 mutant line, RB1004, that is an incomplete knockout (KO) and produces a truncated DBN-1 (1–301 aa)[32]. These experiments revealed an enhanced lifespan in the incomplete KO line compared to the WT strain, N2, upon PQ treatment (10 vs. 9.2 days of median half-life; Supplementary Figure 3d). We surmised that the truncated DBN might be responsible for the observed gain-of-function phenotype. To test this idea, we subjected RB1004 to $dbn-1$ RNA interference (RNAi) to deplete the expression of the truncated DBN and observed now a pronounced reduction of the lifespan under oxidative stress conditions (median half-life of 6.7 days for $dbn-1$-depleted RB1004 vs. 9.2 days for N2 + PQ; Table 1; Supplementary Figure 3e, f and Supplementary Figure 4a). Importantly, $dbn-1$ RNAi treatment of the WT led to a similar reduction in lifespan when subjected to PQ (median half-life of 6.8 vs. 6.7 days of RB1004 + $dbn-1$ RNAi; Supplementary Figure 3f and Table 1). Thus, the residual DBN fragment is interfering with our analyses and we decided to generate a complete $dbn-1$ KO by CRISPR/Cas (JKM1). The absence of $dbn-1$ expression was verified by western blot using an antibody specific for C. elegans DBN-1[32] (Supplementary Figure 4b). The $dbn-1$ KO line showed a similar shortened lifespan upon PQ treatment as RB1004 + $dbn-1$ RNAi or N2 + $dbn-1$ RNAi (median half-life of 6.3 days; Fig. 7a; Table 1). Notably, depletion of $dbn-1$ by either RNAi or the complete KO

led to a reduction of offspring by about 20%, suggesting a systemic effect of $dbn-1$ depletion for the organismal fitness (Fig. 7b). Next, we generated transgenic nematodes that express the adult isoform of human DBN fused to YFP under the control of a pan-neuronal promoter in the ΔDBN-1 background (Δ$dbn-1$/nDBN-YFP, hereafter referred to as nDBN) to create a humanized C. elegans DBN model. We confirmed the expression of human DBN by western blot and the pan-neuronal expression by confocal imaging (Supplementary Figure 3a–c). Human DBN-YFP is expressed throughout the ventral and dorsal nerve cords and can be clearly detected in the head and tail neurons (Supplementary Figure 3a). Lifespan analyses, comparing Δ$dbn-1$ and nDBN nematodes, did not reveal any significant differences under normal growth conditions (Supplementary Figure 3a). However, upon chronic exposure to PQ, nDBN transgenic nematodes exhibited a median half-life of 11.8 days compared to 8.3 days for Δ$dbn-1$ nematodes, suggesting increased stress resistance (Fig. 7c; Table 2).

We previously identified that ROS stimulates ATM-dependent phosphorylation of DBN at serine-647, which accounts for improved stress resilience in neurons. To test the involvement of ATM in DBN function in nematodes, we analysed the impact of ATM inhibition on lifespan using the ATM inhibitor KU55933. Prior to the analysis, we validated the ability of the inhibitor to decrease pS647-DBN in our nDBN C. elegans model. We observed significantly reduced pS647-DBN signals, suggesting that KU55933 is able to inhibit endogenous ATM in C. elegans (Supplementary Figure 4c). Only under conditions of chronic oxidative stress (PQ) did treatment with KU55933 significantly decrease lifespan in nDBN nematodes (half-life of 9.7 days (PQ + KU55933 treatment) vs. 13.7 days (KU55933 treatment) and 11.8 days (PQ treatment), suggesting that loss of ATM function exacerbated the effect of oxidative stress (Fig. 7d–f; Table 2).

Lastly, we investigated if the improvement in lifespan in human DBN-complemented nematodes during sustained oxidative stress involves ATM and DBN-S647 phosphorylation. To test this hypothesis, we generated DBN-1-deficient nematode lines expressing the dephospho-nDBN^S647A or phospho-mimicry nDBN^S647D variants (Supplementary Figure 3a). Notably, nDBN^S647A conferred increased sensitivity to oxidative stress compared to nDBN^S647D (median half-life of 9.7 vs. 11.8 days; Fig. 7e; Table 2 and Supplementary Figure 4d). Since both phosphorylation mutants alter lifespan during chronic oxidative stress independent of ATM function, DBN-S647 presents an

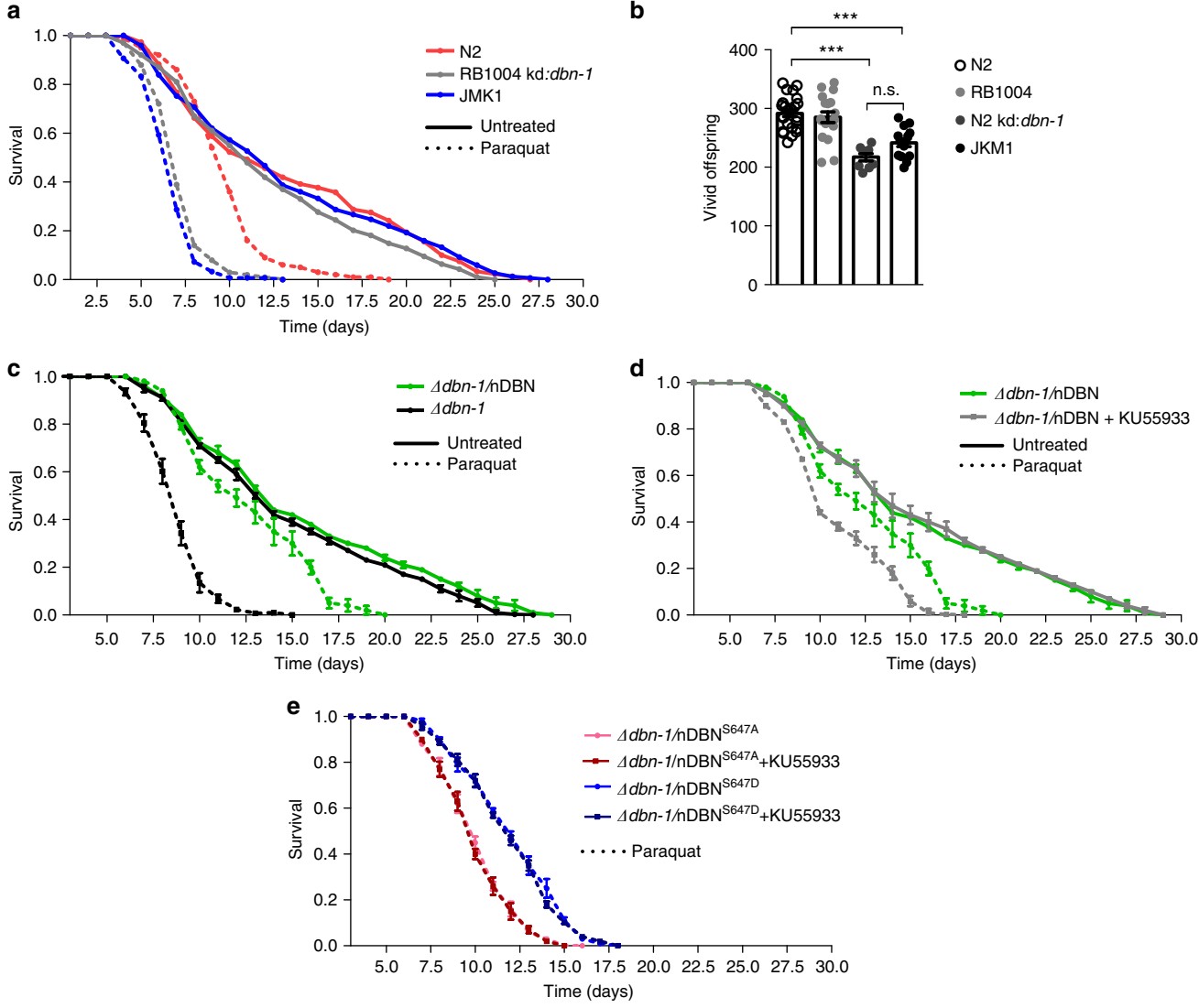

**Fig. 7** Serine-647 DBN regulates lifespan and stress resistance. The lifespan of *C. elegans* was analysed in DBN-1 deficient (Δ*dbn-1*) nematodes expressing human DBN in the absence or presence of oxidative stress using paraquat and/or ATM inhibitor (KU55933). *N* = number of cohorts, *n* = number of worms. **a** Cumulative survival probability of N2 (red), *dbn-1* RNAi-treated RB1004 (grey) and JKM1 (blue), treated with (dashed lines, N2: *N* = 1, *n* = 101; RB1004 kd:*dbn-1*: *N* = 1, *n* = 100; JMK1: *n* = 1, *n* = 150) or without paraquat (continuous lines, N2: *N* = 1, *n* = 150; RB1004 kd:*dbn-1*: *N* = 1, *n* = 100; JMK1: *N* = 1, *n* = 150). **b** Progeny assay showing vivid offspring per parental nematode. Ten to twenty-five nematodes were analysed per strain. *dbn-1* RNAi-treated N2 and JKM1 differ significantly from untreated N2 and RB1004. Mean ± SEM. One-way ANOVA Bonferroni test, *** *p* < 0,001. n.s.: non-significant. N2: *N* = 1, *n* = 24; RB1004: *N* = 1, *n* = 18; N2 kd:*dbn-1*: *N* = 1, n = 9; JKM1: *N* = 1, *n* = 15. **c** Cumulative survival probability of Δ*dbn-1* (black lines) or Δ*dbn-1* expressing nDBN (green lines) in the absence (continuous lines) or presence (dashed lines) of paraquat. Mean ± SEM. Δ*dbn-1*: *N* = 3, untreated: *n* = 310, paraquat: *n* = 290. Δ*dbn-1*/nDBN: *N* = 3, untreated *n* = 315, paraquat, *n* = 300. **d** Cumulative survival probability of Δ*dbn-1* expressing nDBN treated with the ATM inhibitor KU55933 (grey line) or DMSO-control (green line) for 22 h before the start of the assay in the presence or absence of paraquat. Mean ± SEM. Δ*dbn-1*/nDBN: *N* = 3, untreated *n* = 315, paraquat *n* = 300; Δ*dbn-1*/nDBN + KU55933: *N* = 3, *n* = 300. **e** Cumulative survival probability of Δ*dbn-1* expressing human dephospho (nDBN^S647A, blue lines) or phospho-mimicry (nDBN^S647D, red lines) mutants treated with or without ATM inhibitor (KU55933) in the presence or absence of paraquat. Mean ± SEM. Δ*dbn-1*/nDBN^S647A: *N* = 3, untreated: *n* = 315, +KU55933: *n* = 300; Δ*dbn-1*/nDBN^S647D: *N* = 3, untreated: *n* = 290, +KU55933: *n* = 300. DBN developmentally regulated brain protein, ATM ataxia-telangiectasia mutated, DMSO dimethyl sulphoxide, ANOVA analysis of variance

important ATM substrate in this response. In the absence of ATM function, the protective effect of the DBN^S647D mutant in response to sustained oxidative stress may be due to different protein levels. Indeed, our quantification revealed an expression of nDBN^S647D of about 25% of the WT nDBN (Supplementary Figure 3b, c see Methods section). However, the fact that the treatment of WT nDBN with the ATM inhibitor reduced the median half-life to levels detected in the less expressed DBN^S647A mutant (median half-life of 9.7 days for WT + ATM inhibitor vs.

9.6 days for nDBN^S647A) questions a dose-dependent effect and substantiates the role of phosphorylation in mediating oxidation resistance. Taken together, our results demonstrate that DBN phosphorylation on S647 is an integral mechanism involved in DBN-mediated stress tolerance to oxidative stress (Fig. 8).

## Discussion
The major finding from this study is the identification of DBN as an ATM substrate and of a mechanism to protect against

oxidative stress-induced cellular dysfunction. During sustained, energy-demanding cellular activities, the coordination of cytoskeletal remodelling by ATM and DBN may be key to protecting against generated ROS. Given that filamentous actin has been suggested to be selectively lost from synapses during the early stages of AD[33], DBN and ATM may regulate actin organization in concert, and thereby safeguard against early synaptic dysfunction and cognitive decline. Exploiting cytoskeletal dynamics to withstand the proximity of energy consumption and oxidation stress may be a fundamental principle that is relevant during ageing, and also during developmental growth and regenerative processes[27,34,35].

## Methods

All animals used were handled in accordance with the relevant guidelines and regulations. Protocols were approved by the 'Landesamt für Gesundheit und Soziales' (LaGeSo; Regional Office for Health and Social Affairs) in Berlin and animals reported under the permit number T0347/11.

**Aβ$_{1-42}$ preparation**. Oligomeric Aβ$_{1-42}$ were prepared essentially as described by Klein[12,34]. Briefly, Aβ peptide (Peptide Synthesis Core Facility, Charité Berlin) was dissolved in ice-cold 1,1,1,3,3,3-hexafluoro-2-propanol (HFIP, Sigma) at a concentration of 2.5 mg/ml and stored in 50 µl aliquots at −80 °C. For oligomeric preparations, HFIP was evaporated in a SpeedVac for 30 min. and the monomerised peptide was re-solubilized in 5.5 µl of dimethyl sulphoxide. The peptide was then diluted into Dulbecco's modified Eagle's medium (DMEM)/F12 (Thermo Fisher) to a concentration of 100 µM and incubated at 4 °C for 48 h to form oligomers. Tubes were finally centrifuged at 17,000 ×g for 15 min to remove potential filamentous material and the supernatant containing the oligomer preparation was transferred to a new tube. Final dilutions between 10 and 0.01 µM were prepared in Neurobasal medium and applied to hippocampal cultures for 24 h. Due to the heterogeneity in Aβ$_{1-42}$ assembling states in the soluble preparation[34], we refer to the molar concentrations of Aβ$_{1-42}$ based on the starting Aβ$_{1-42}$ peptide. Concentrations below 1 µM did not result in any detectable spine density reduction and concentrations above 1 µM resulted in dendrite degeneration.

**Mice**. DBN KO mice were generated as previously described[10]. Atm+/− mouse was generated by Anthony Wynshaw-Boris lab[31] and kindly provided by Clemens Schmitt (Charité, Berlin). In order to obtain ATM KO neurons, E15 embryos were removed from Atm+/− pregnant mice, genotyped and neurons from Atm+/+ and Atm−/− embryos were prepared as described in the following section.

**Mouse hippocampal culture**. Hippocampi were dissected from E16.5 C57BL/6 mice embryos and dissociated using 0.5 mg/ml trypsin in Hank's balanced salt solution (HBSS) for 15 min at 37 °C. Five washing steps were performed, two in HBSS buffer, one in growth medium (Neurobasal medium supplemented with 1% Glutamax, 2% B27, 1% penicillin/streptomycin and 100 µM β-mercaptoethanol) and one final washing step in Neurobasal A supplemented with 10% horse serum. Tissue pieces were then triturated in growth medium using fire polished Pasteur pipettes. Single-cell suspension cells were plated on glass coverslips (Karl Hecht) previously washed with 100% methanol, 70% ethanol and 100% ethanol and coated with 30 ng/µl poly-ornithine. For biochemical experiments, neurons were plated in 6-well plates at a density of 400,000 cells per well. For cellular imaging, neurons were plated in a 12-well plate at 100,000 cells per well. Neurons were maintained in a humidified atmosphere at 37 °C and 5% CO$_2$ for the indicated amount of time.

**Rat neuron culture**. Rat hippocampi and cortices were dissected from Wistar rats postnatal stages (P0–P1) and dissociated in papain (Worthington) according to the manufacturer's protocol. Twenty thousand hippocampal cells were plated in growth medium (Neurobasal medium supplemented with 1% Glutamax, 1% B27) on 35 mm glass-bottom dishes (MatTek Corporation) previously coated with 30 ng/µl poly-D-lysine. Two hours after plating (30,000 cells per dish), 700 µl of rat-conditioned medium (80% growth medium, 15% glial medium and 5% cortical conditioned medium) was added to every dish until used for FUNCAT-PLA experiments (days in vitro (DIV) 17–24). For cortical cultures, single-cell suspension was plated in growth medium on T75 flasks previously coated with 30 ng/µl of poly-ornithine. The medium was collected 10 days later. Rat glial cultures were prepared using single-cell suspension of dissociated cortices plated in minimum essential medium supplemented with 10% horse serum, 20% glucose and 1% penicillin/streptomycin on T75 flasks previously coated with 0.5% collagen. When cell confluence was achieved, the medium was replaced with growth medium and collected every 3 days.

**Constructs**. YFP-DBN and FLAG-DBN constructs were generated in the Eickholt lab and are based on the human cDNA sequence (Homo sapiens DBN1, transcript variant 1, NCBI Reference Sequence: NM_004395.3 [https://www.ncbi.nlm.nih.gov/nuccore/NM_004395.3] corresponding to DBN E)[22]. DBN constructs with amino acid substitutions were generated by site-directed mutagenesis and validated by sequencing. pET-28b-His-DBN-Intein was generated by PCR amplification from DBN-YFP using primers (DBN-Nde-5′-ATACATATGGCCGGCGTCAGCT TCAGC-3′ and DBN-Intein-Xho-5′-GTACTCGAGATCACCACCCTCGAAGCC CTC-3′) creating flanking restrictions sites for NdeI and XhoI. The amplified DBN cDNA was inserted into the pJET cloning vector (Thermo Scientific), where, subsequently, the internal NdeI site was silently mutated through a standard site-directed mutagenesis protocol (DBN-NdeI-BsrGImut-5′- CGGCCGACTGGGC TCTGTATACCTATGAAGATGGCTCCGATG-3′ and DBN-NdeI-BsrGImut-5′-CATCGGAGCCATCTTCATAGGTATACAGAGCCCAGTCGGCCG-3′).The generated cDNA was then cloned into pMXB10 (NEB), in frame with a sequence encoding a cleavable chitin-binding domain (Intein). To enable efficient protein expression, the DBN-Intein cDNA was transferred into pET-28b (Novagen) in frame with the His-tag of the multiple cloning site using NdeI/BamHI. roGFP1 was amplified from pDONR221-roGFP1[36] using primers (roGFP F-5′-GTGGATC CACCGGTCGCCACCATGGTGAGCAAGGGCGAGGAGCTGT-3′ and roGFP R-5′-GAGCGGCCGCTTTACTTGTACAGCTCGTCCATGCCGAGAGTGA-3′) inserting NotI and BamH1 restriction sites. The amplified product was then cloned in frame behind the largest human DBN isoform (DBN1 iso3 [https://www.uniprot.org/uniprot/Q16643UniProtKB], Q16643-3 corresponding to DBN A) to obtain pDBN-roGFP1. pcDNA3.1(+)FLAG-His-ATM cDNA was a gift from Michael Kastan (Addgene plasmid # 31985).

**Culture and transfection of HEK 293T cells and COS-7 cells**. HEK 293T (ATCC, CRL-11268) and COS-7 (ATCC, CRL-1651) cells were cultured in DMEM supplemented with 10% foetal calf serum and 1% penicillin/streptomycin at 37 °C and 5% CO$_2$. When required, cells were plated on 15 ng/µl poly-ornithine-coated plates and transfected using Lipofectamine 2000 (Invitrogen) according to the manufacturer's

### Table 1 Comparing the median lifespan of N2, RB1004 and JKM1 upon *dbn-1* RNAi

|  | **N2** | **RB1004** | **JMK1** |
|---|---|---|---|
| Nontreated | 11 | 13.4 | 11.4 |
| kd:dbn-1 | 12.7 | 10.7 | – |
| +paraquat | 9.2 | 10 | 6.3 |
| kd:dbn-1 + paraquat | 6.8 | 6.7 | – |

Treatment with *dbn-1* RNAi with or without paraquat-induced oxidative stress. (–) No siRNA treatment was performed with JKM1 (complete knockout)
*RNAi* RNA interference, *siRNA* small interfering RNA

### Table 2 Median half-life data of Δ*dbn-1*, nDBN, nDBNS647A or nDBNS647D treated with paraquat and/or ATM inhibitor (KU55933)

|  | **No DBN** RB1004 kd:*dbn-1* | +nDBN variants | | |
|---|---|---|---|---|
|  |  | **wt** JKM1 kd:*dbn-1* | **S647A** RB1004 kd:*dbn-1* | **S647D** RB1004 kd:*dbn-1* |
| Non-treated | 13.3 | 13.3 | 13 | 14 |
| +KU55933 | 13.7 | 13.7 | 13.1 | 15.5 |
| +paraquat | 8.3 | 11.8 | 9.7 | 11.8 |
| +paraquat + KU55933 | 11 | 9.7 | 9.6 | 11.7 |

*ATM* ataxia-telangiectasia mutated, *DBN* developmentally regulated brain

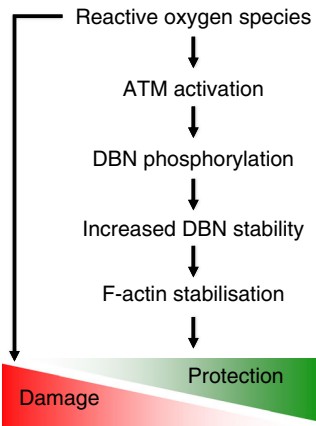

**Fig. 8** Proposed odel. Increased oxidative stress is counteracted by a mechanism in which activation of ATM phosphorylates DBN, which in turn stabilizes DBN and prevents actin depolymerization. ATM ataxia-telangiectasia mutated, DBN developmentally regulated brain protein

requirements. At 4 h after transfection, the medium was replaced with fresh medium, and cells were incubated at 37 °C and 5% $CO_2$ for further analysis.

**Viral transduction**. For viral transduction, lentiviral vectors based on pLenti6/V5-D-TOPO (Invitrogen) expressing YFP, DBN-YFP, DBN(S647A)-YFP, DBN(S647D)-YFP (numbering of amino acids refers to the largest human isoform, DBN1 iso3) or DBN-roGFP under the control of the synapsin 1 promoter (to ensure neuronal expression) were cloned and viral particles produced by the viral core facility (Charité Berlin). Cultures were infected with a multiplicity of infection of 0.5 at 7 DIV (for DBN-ro-GFP) or 14 DIV for all other transgenes.

**Immunocytochemistry**. Neuronal cultures for dendritic spine density analysis were fixed in 4% paraformaldehyde (PFA), 4% sucrose in a cytoskeleton-stabilizing PHEM buffer (PIPES, HEPES, EDTA and $MgCl_2$, pH 7.4) for 30 min. Neurons were permeabilized in PHEM with 0.1% Triton for 10 min and blocked in PHEM with 4% goat serum. Dendrites were stained using MAP2 (mouse) for 1 h followed by F-actin staining using Phalloidin-conjugated fluorescent dye Acti-stain 488 or Acti-stain 670 phalloidin for 1 h (Cytoskeleton).

**Immunohistochemistry**. Adult mice were sedated with isoflurane, perfused and sacrificed. Coronal brain slices with 60 μm diameter were obtained by cryotome sectioning and subsequently permeabilised with 2% Triton X-100 in phosphate-buffered saline (PBS) for 24 h. Slices were incubated with primary antibodies for 48 h in PBS with 0.1% Tween-20 (PBS-T). After three washes with PBS-T, slices were incubated with species-cross-absorbed secondary antibodies in PBS-T for a further 48 h. As preparation for tissue clearing, slices were fixed after three washes with PBS in 4% formaldehyde (weight per volume) in PBS for 1 h at 4 °C. Tissue clearance was performed using ScaleA2 (4 M urea, 10% (weight per volume) glycerol, 0.1% (weight per volume) Triton X-100 and 0.1× PBS) for 3 days, as described previously[37]. Transparent slices were mounted onto glass slides using Mowiol with 4 M urea. Slices were analysed using a Leica Sp8 inverse confocal microscope equipped with a ×63 objective (NA 1.7), whereas the channels were separately acquired by a sequential scanning mode. The correlation between the intensity of DBN and the intensity of pS647-DBN was analysed by tracing four random lines across the image and performing a plot profile to obtain the fluorescent intensity at each pixel.

**FUNCAT-PLA**. FUNCAT-PLA was performed as previously described[21]. Briefly, hippocampal neurons were incubated for 2 h with Neurobasal A without methionine supplemented with 1% Glutamax, 2% B27 and 4 mM AHA. In some experiments, the medium was additionally supplied with 40 μM anisomycin or 4 mM methionine. After metabolic labelling with AHA, neurons were washed twice in PBS (pH 7.4), supplemented with 1 mM magnesium, 0.1 mM $CaCl_2$ (PBS-MC) and fixed with 4% PFA, 4% sucrose in PBS-MC for 20 min. Neurons were washed in PBS-MC, permeabilised for 15 min in 0.5% Triton and blocked for 1 h in 4% goat serum. A biotin-alkyne click reaction (Click-iT® Protein Reaction Buffer Kit, Invitrogen) was then performed as suggested in the manufacturer's protocol. After a step of permeabilisation in 0.5% Triton and a blocking step in 4% goat serum, cells were labelled with anti-biotin and anti-DBN antibodies and a proximity ligation assay was performed using Duolink reagents (Sigma). Rabbit PLA^plus and mouse PLA^minus probes were used for secondary antibodies and Duolink Detection reagents for ligation, amplification and label probe binding. The dendrites were

stained with MAP2 (guinea pig) and the nucleus with Hoechst. Imaging was performed using either a Confocal Laser Scanning Microscope Leica TCS SP8 using a ×63 oil objective or a laser scanning microscope 780 (LSM 780) confocal microscope (Zeiss) using a ×40 oil objective. Images were acquired with a resolution of 1024 × 1024 pixels through the entire sample as z-stacks size 0.5 μm. Laser intensities and gain were defined for every experiment and maintained without changes within an experiment. Five to ten images per condition were captured and processed for analysis in Fiji. All the images obtained from the FUNCAT-PLA experiments were performed on maximal projections using a semi-automated Plugin adapted from tom Dieck et al.[21]. Briefly the PLA signal overlapping or in close proximity (<1 μm) with the MAP2 signal was normalized to the MAP2 area to take in account the neuron size and elaborate structure.

**Live cell imaging**. COS cells were grown on glass IBIDI dishes in DMEM, 10% FCS, 1% penicillin/streptomycin, transfected with DBN-roGFP using Lipofectamine and imaged 24 h later. Hippocampal neurons were cultured in Neurobasal A medium supplemented with 1% Glutamax, 2% B27, 1% penicillin/streptomycin on IBIDI glass coverslip. Neurons were infected with DBN-roGFP on DIV7 and imaged on DIV21.

Dishes were transferred to a 37 °C live cell incubator supplied with 5% $CO_2$ for microscopy and imaged on a Nikon spinning disk confocal CSU-X. When required, 100 μM of APV was applied at least 1 h prior to measurements. Cells were excited with 405 and 488 nm lasers. Fluorescence emission was detected with a narrow 510/520 filter using iXon3 DU-888 Ultra camera (Andor). COS cell images were acquired every 10 s using a ×40 objective (NA 1.3) for 400 s and hippocampal neurons were imaged every 5 s using a ×60 objective (NA 1.4) for 255–280 s. In COS cells, $H_2O_2$ was used at a concentration of 0.1 mM and DTT at 7 mM. In hippocampal cells, $H_2O_2$ was used at 10 mM, DTT at 7 mM and NMDA at 50 μM. Individual COS cells or regions of interest drawn around individual dendritic spines chosen randomly throughout the neuron were analysed using Fiji, and the ratio of emission intensities excited at 405 and 488 nm were measured.

**Spine density analysis**. For the detection and density analysis of dendritic spines, confocal images of hippocampal neurons at 21 DIV labelled with Acti-stain (Phalloidin) were taken on a Leica TCS SP8 microscope (constant parameters: laser line 552: 1.43% power; HyD1 detector, gain: 33%) and analysed by the Imaris software (Bitplane, version 8.1.2). Starting with a 3-D image of a whole neuron, the software's filament tool detected a dendritic tree based on the MAP2 signal and predefined parameters using automated seed point and intensity thresholds and a maximal gap length fixed to 15 μm. Manual editing was then used to remove stray branches, if necessary. Spines were subsequently detected as protrusions of a predefined size range along the identified dendrite based on Acti-stain 488 Phalloidin signal threshold 40.0 (Fig. 1a). To shape the spines, we used one of the two optional algorithms ('shortest distance from distance map'). By measuring total dendritic length and number of spines, the mean spine density/10 μm along the entire neuron's dendrites was calculated. In addition, we analysed the spine head centre voxel Phalloidin (F-actin) intensity as an indicator of F-actin stability in the spine centre.

**Protein lysate preparation, SDS-PAGE and western blotting**. Whole rat or mouse brains were weighed and homogenized in 4× the volume of RIPA buffer (50 mM Tris-HCl, pH 7.4, 150 mM NaCl, 0.5% sodium deoxycholate, 1% NP40, 0.1% sodium dodecyl sulphate (SDS)) supplemented with protease inhibitors (Calbiochem set III) and phosphatase inhibitors (1 mM $Na_2MO_4$, 1 mM NaF, 20 mM β-glycerophosphate, 1 mM $Na_3VO_4$, 500 nM cantharidin) using a glass pestil. Homogenates were centrifuged at 20,000 × *g* and the supernatant was collected for further protein quantification analysis using BCA Thermo Scientific Pierce™ Protein Assay. Cells (neurons or 293T) were washed once with cold PBS and lysed in cold RIPA buffer or otherwise specified, supplemented with protease inhibitors and phosphatase inhibitors. Cell lysates were centrifuged at 20,000 × *g* and the supernatant was transferred to a tube containing Roti load I sample buffer. In average 15–30 μg of protein was loaded on an SDS-polyacrylamide gel electrophoresis (SDS-PAGE) gel. ATM and pATM was detected using 80 μg of protein. Western blot analysis was performed as previously described[38]. Quantification of band densities was performed using Fiji. The area of the band and the mean grey value were measured to obtain a relative density. For relative quantifications, measurements were normalized to loading control.

**Antibodies**. Anti-DBN M2F6 (1:1000 for western blot, 1:100 for immunohistochemistry, ADI-NBA-110-E Enzo), anti-ATM 2C1(1A1) (1:200, ab78 Abcam), anti-pS1981-ATM 10H11.E12 (1:400, 200-301-400S, Rockland), anti-p-p44-42 ERK (pMAPK, 1:1000 #9101, CST), anti-α-tubulin (1:5000, T6199, Sigma), anti-GAPDH (1:5000, CB1001, Merck), anti-pS15-p53 (1:1000 #9284, CST), anti-poly-monoubiquitin FK2 (1:1000, BML-PW8810, Enzo), anti-HA 3F10 (1:1000, 11867423001, Roche), anti-FLAG M2 (1:10,000, F3165, Sigma), anti-cyclin B1 (1:1000, sc-245, Santa Cruz), anti-MAP2 mouse (1:500 for immunocytochemistry M9942, Sigma), anti-MAP2 guinea pig (1:1000 for FUNCAT-PLA immunocytochemistry or 1:500 for immunohistochemistry, 188 004 Synaptic System), anti-CaMKII (1:250, ab92332 Abcam), anti-pS647-DBN (1:5000 for western blot, 1:250

for immunohistochemistry, numbering of amino acids refers to the largest human DBN isoform (DBN1 iso3[22], anti-pS142-DBN[17] (1:500), anti-DBN-1 *C. elegans* specific (1:500)[32], anti-biotin mouse (1:5000, B7653 Sigma) or anti-biotin rabbit (1:5000, #5597 Cell Signalling). The most important uncropped western blots are included in the Supplementary information.

**Production of recombinant DBN.** Recombinant His-DBN-Intein was produced using the Chitin-Intein system IMPACT™ (Intein Mediated Purification with an Affinity Chitin-binding Tag, NEB). Briefly, Rosetta DE3 bacteria were transformed with pET-28b-His-DBN-Intein and the expression was induced for 3 h at 20 °C using 1 mM isopropyl β-D-1-thiogalactopyranoside (IPTG). Bacteria were lysed in HEPES lysis buffer (20 mM Tris-HCl, pH 7.5, 50 mM NaCl, 1 mM tris(2-carboxy-ethyl)phosphine (TCEP), 4-(2-aminoethyl) benzenesulphonyl fluoride hydrochloride (AEBSF), pepstatin A, leupeptin), sonicated and centrifuged at $30,000 \times g$ for 30G min. Chitin beads (NEB) were incubated with lysates for 1 h at 4 °C followed by several washing steps with 20 mM Tris-HCl (pH 7.5), 500 mM NaCl, 1 mM TCEP, AEBSF, pepstatin A and leupeptin.

**Kinase assay.** Approximately 5 μg of freshly produced Intein-DBN was used in each assay. For the kinase assays using brain lysates, 100 μg of adult mouse brain extracted in 50 mM Tris-HCl (pH 7.5), 150 mM NaCl, 0.1% Triton, protease inhibitors (Calbiochem set III) and phosphatase inhibitors ($Na_3MO_4$, NaF B-glycerophosphate, $Na_3VO_4$, cantharidin) was used per reaction. When required, 10 μM KU55933 was added to the brain extracts before the start of the reaction. The kinase assay was performed at 37 °C for 1 h in 25 mM HEPES, 75 mM NaCl, 3 mM $MgCl_2$, 2 mM $MnCl_2$, 100 μM ATP with or without 10 μM KU55933. For the kinase assays using FLAG-ATM, HEK 293T cells were plated on poly-ornithine-coated T75 flasks and transfected with FLAG-ATM using Lipofectamine. After 48 h, cells were treated with 250 μM $H_2O_2$ for 90 min before lysis in ATM lysis buffer (20 mM HEPES, 150 mM NaCl, 0.2% Tween, 1.5 mM $MgCl_2$, 1 mM EGTA, 0.25 mM DTT and benzonase 1/1000). Following a preclearing step using protein G agarose for 1 h at 4 °C, FLAG-ATM was immunoprecipitated with anti-FLAG M2 affinity gel (Sigma) for 2 h. Beads were washed 3× 15 min with ATM lysis buffer followed by 2× 15 min washes in kinase buffer (25 mM HEPES, 75 mM NaCl, 3 mM $MgCl_2$, 2 mM $MnCl_2$). Recombinant Intein-DBN was incubated with FLAG-ATM immunoprecipitates for 10 min at 37 °C in kinase buffer before the addition of 50 μM ATP for an extra hour. The reaction was stopped by the addition of Roti load I (Ruth).

**Ubiquitination assay.** HEK 293T cells were plated on poly-ornithine-coated 6-well plates. At 24 h after plating, cells were transfected with HA-ubiquitin-GFP and 48 h after plating with FLAG-DBN. Ten micromoles of MG132 was added 72 h after plating for 5 h. Cells were lysed in RIPA buffer, NEM 25 mM and MG132 10 μM. FLAG-DBN was immunoprecipitated with anti-FLAG M2 affinity gel (Sigma) previously blocked in 3% bovine serum albumin, 5% milk, 1% goat serum for 1 h and precleared with protein A/G agarose (Pierce) for 40 min. Immunoprecipitates were washed three times in RIPA buffer and three times in high salt wash buffer (50 mM Tris-HCl, pH 8, 350 mM NaCl, 1% NP40, 0.5% sodium deoxycholate, 0.1% SDS).

**Metabolic labelling and pulse-chase experiments.** HEK 293T cells were plated on poly-ornithine-coated 12-well plates. At 24 h after transfection, cells were washed and incubated for 1 h in pre-warmed methionine-free DMEM supplemented with 10% FCS and 1% penicillin/streptomycin. Pulse labelling for 1 h was performed using 1 mM AHA in methionine-free DMEM. After the pulse, cells were washed once in complete medium and twice in PBS, followed by further chase in complete medium for 24, 48 or 72 h before cell lysis. When inhibiting the proteasome, 1 μM of MG132 was added to the complete chase medium.

Cell lysis was performed in 150 μl RIPA buffer supplemented with protease inhibitor cocktail and phosphatase inhibitors. One hundred microliters of cell lysates were collected and click chemistry was performed using the Click-iT® Protein Reaction Buffer Kit (Invitrogen) as suggested in the manufacturer's protocol. Cycloaddition was used with biotin-alkyne and precipitated protein was resuspended in 1× Roti load sample buffer for western blot analysis. Labelled proteins were detected with streptavidin-horse radish peroxidase (Cell Signalling) and later stripped for 30 min at 45 °C to blot further with anti-FLAG. The protein exponential decay was achieved considering that at time point 0 h chase 100% labelling is achieved. Protein half-life was thereafter calculated for each experiment and later mean values between half-life was obtained ($n \geq 3$).

**Caenorhabditis elegans strains.** The strains N2, WT and RB1004, *ok925* (Δ*dbn-1* [Exon 5/6]) were provided by the Caenorhabditis Genetics Centre, which is funded by NIH Office of Research Infrastructure Programs (P40 OD010440). The partial deletion in *dbn-1* gene of RB1004 results in the expression of a truncated DBN-1 protein[32]. Complete depletion of DBN was achieved by additionally using RNAi (L4440-*dbn-1*, Ahringer library). Alternatively, we also generated a new strain JKM1, Δ*dbn-1* [Exon 1–6], where exons 1 to 6 were deleted by CRISPR/Cas9[39,40]

using four different guide RNA sequences (sgRNA1: 5′-CAACTATCATGCACAC GGCA-3′; sgRNA4: 5′-TAGCGGGAAAAAGGCCTACG-3′; sgRNA5: 5′-GATGG CTCTCTGGGACTACC-3′; sgRNA7: 5′-AGGAGATTGAAGCGTCGTAT-3′). pM B67 (50 ng/μl) (*hsp-16.48promoter::Cas9*) (Boxem Lab), 50 ng/μl pJJR50-sgRNA1 (containing 'sgRNA1' under control of U6 promoter), 50 ng/μl pJJR50-sgRNA4 (containing 'sgRNA4' under control of U6 promoter), 50 ng/μl pJJR50-sgRNA5 (containing 'sgRNA5' under control of U6 promoter), 50 ng/μl pJJR50-sgRNA7 (containing 'sgRNA7' under control of U6 promoter)[39], 5 ng/μl *myo-2promoter*:: mCherry (obtained from Morimoto Lab), 3 ng/μl pIR98 (*rps-0promoter::hygR CeOpt::unc-54′UTR*) (Chin Lab) were mixed and microinjected into 40 young adult N2 nematodes. After detection of the red fluorescent pharyngeal marker in the offspring, 750 μg of hygromycin B (Sigma) was added to the plates. Nematodes were grown until several young adults were present. These nematodes were heat shocked for 2 h at 34 °C. After 12 h incubation at 20 °C, the nematodes were transferred onto a fresh plate to lay eggs. After 6 h of egg laying, the adult animals were removed and 750 μg hygromycin B was added to the plate. This procedure was repeated two times. After the final third heat shock, 20 nematodes were singled and genotyped with primers binding upstream and downstream of the *dbn-1* locus (forward: 5′-GCCGCTCAACTACCGTAACT-3′; reverse: 5′-CAGGAAGTGG-GAGAATGGGAG-3′; WT fragment 3315 bp; KO fragment 900–1100 bp).

Once depletion of DBN-1 was achieved, three new strains were generated where human DBN$^{wt}$-YFP, DBN$^{S647A}$-YFP or DBN$^{S647D}$-YFP were specifically expressed in neuronal tissue using the *rgef-1* promoter. The strains nDBN$^{S647A}$-YFP [*rgef-1promoter*::hDBN(S647A)-YFP::*unc-54′UTR* + *myo-2promoter*::mCherry::*unc-54′UTR* + *ok925*(Δ*dbn-1* [Exon 5/6])] and nDBN$^{S647D}$-YFP [*rgef-1promoter*::hDBN(S647D)-YFP::*unc-54′UTR* + *myo-2promoter*::mCherry::*unc-54′UTR* + *ok925*(Δ*dbn-1* [Exon 5/6])] were generated by ballistic transformation of strain RB1004. Extrachromosomal arrays have been integrated into genomic DNA by γ-irradiation. Strains were back-crossed four times with RB1004 to eliminate background mutations. Analysis of the expression levels of nDBN and phospho-mutants revealed differential expression, which may be due to variations of integration site and copy number of the transgenes into the *C. elegans* genome, and also due to their stability on messenger RNA and protein level. The strain nDBN(wt)-YFP, *rgef-1promoter*::hDBN(wt)-YFP::*unc-54′UTR* + Δ*dbn-1* [Exon 1–6], was generated by microinjection of 30 ng/μl pPD95-77_nDBN-YFP(wt) (*rgef-1promoter*::hDBN (wt)-YFP::*unc-54′UTR*) + 100 ng/μl short DNA fragments (GeneRuler 50 bp DNA Ladder, Thermo) into strain JKM1. Extra-chromosomal arrays have been integrated into genomic DNA by γ-irradiation. The strain was back-crossed four times with JKM1 to eliminate background mutations.

**Maintenance of *C. elegans*.** Nematodes were maintained at 20 °C on nematode growth medium (NGM) using established protocols[36]. Culture dish size was 60 mm, and filled with 8 to 10 ml NGM. RNAi plates contained in addition 1 mM ampicillin and 1 mM IPTG.

**Lifespan assay of *C. elegans*.** To ensure correct DBN knockdown and this before the start of the assay, nematodes were maintained on RNAi plates for two generations with HT115 bacteria containing L4440-*dbn-1* (Ahringer library) as food source to knockdown *dbn-1*. Nematodes were age synchronized by bleaching. Eggs were starved for 22 h in M9 medium (8.5 mM NaCl, 56 mM $Na_2HPO_4$, 14 mM $K_2HPO_4$) at 20 °C. Hatched L1 larvae (day 1) were placed on RNAi plates and kept at 20 °C until they reached L4 stage (day 3). The L4 larvae were washed off the plates with M9 medium, washed once with S-Basal (100 mM NaCl, 6 mM $K_2HPO_4$, 44 mM $KH_2PO_4$, 5 mg/l cholesterol) and pelleted on ice. Approximately 150 washed nematodes were transferred into wells of a 12-well plate and topped up with prepared S-Basal (supplemented with 30 mM $MgSO_4$, 30 mM $CaCl_2$, 15 μg/ml cholesterol, $2 \times 10^7$ bacteria/ml (HT115-L4440-*dbn-1*)) to a final volume of 3 ml. When required S-Basal was supplemented with 500 μM KU55933 (Sigma) or 1% (v/v) dimethyl sulphoxide as a solvent control. After incubation for 48 h at 20 °C, the nematodes were transferred to RNAi plates containing 2 mM PQ (Sigma) or control RNAi plates without PQ. Nematodes were cultured at 20 °C and scored every day. Survival was tested by gently prodding nematodes with a platinum wire. The nematodes were transferred to fresh plates every day until day 10. From then on, they were transferred every 3 days. Lifespan assays were repeated three times with a cohort size of approximately 100 worms.

**Preparation of *C. elegans* lysate for western blotting.** Full grown plates of nematodes were harvested by rinsing them off with M9 medium. Nematodes were pelleted on ice and washed further with M9 three times. For the analysis of human DBN protein levels, fresh nematode pellets were lysed in RIPA buffer (50 mM Tris-HCl, pH 7.4, 150 mM NaCl, 0.5% sodium deoxycholate, 1% NP40, 0.1% SDS) supplemented with protease inhibitors (Calbiochem set III, AEBSF, Sigma 1 mg/ml), benzonase (Millipore, 1:100) and phosphatase inhibitors (1 mM $Na_2MO_4$,1 mM NaF, 20 mM β-glycerophosphate, 1 mM $Na_3V$)4), 500 nM cantharidin) using a tissue homogenizer (Minilys, Bertin Instruments). Homogenates were centrifuged at $20,000 \times g$ and supernatant was collected for further protein quantification analysis using BCA Thermo Scientific Pierce™ Protein Assay. Forty micrograms of protein was loaded on an SDS-PAGE and further analysed by western blotting. For

the analysis of the expression level of *C. elegans* DBN-1, nematode pellets were resuspended in 2× sample loading buffer (50 mM Tris-HCl, pH 6.8, 2% SDS, 15% glycerol, 0.1% bromophenol blue, 100 mM DTT) and incubated for 5 min at 98 °C. Samples were centrifuged for 5 min at 5000× *g* and the supernatant was loaded on a 10% SDS-PAGE.

**Confocal imaging of *C. elegans*.** Adult nematodes (day 4/5) expressing nDBN^wt^-YFP, nDBN^S647A^-YFP and nDBN^S647D^-YFP were anaesthetised with 2 mM levamisole (levamisole hydrochloride) (Sigma) and immobilized on a 3% agarose pad.

Imaging was performed on a Zeiss confocal LSM 780. Fluorophores were excited at 488 nm and emission was measured from 490 to 581 nm. Used objectives were 'EC Plan-Neofluar ×10/0.30 M27' and 'Plan-Apochromat ×20/0.8 M27'.

**Fluorescence quantification of DBN-YFP in *C. elegans*.** Adult nematodes (day 4) expressing nDBN^wt^-YFP, nDBN^S647A^-YFP and nDBN^S647D^-YFP were anaesthetised with 2 mM levamisole (levamisole hydrochloride) (Sigma) and immobilized on a 3% agarose pad. Images were recorded on a spinning disk microscope (Axiovert 200M, Carl Zeiss; Perkin Elmer System with Yokogawa spinning disk, CSU22; EMCCD camera, Hamamatsu C9100-50). Magnification was ×400. Z-stacks of nematode heads were taken from the very top to the very bottom of the head (approx. 28–32 μm) in 1 μm steps. Field of view was set such that the head fit from the tip of the mouth to the retrovesicular ganglion. Exposure time was set to 75 ms for every image of the stacks. Laser power and sensor sensitivity were kept constant. Images were analysed using Fiji. First a 'Sum slices Z-Projection' was performed to get one single layer image from the raw data. The fluorescence intensity was measured and background fluorescence was subtracted.

**Statistical analysis.** In the FUNCAT-PLA experiments, to plot the data obtained from the pulse-chase experiments, we reasoned that at the time point 0 h chase, maximum protein labelling has been achieved. Therefore, we normalized the data considering AHA 0 h as 100%. *N* represents the number of independent experiments and *n* the number of single neurons. In each condition, $n = 5$–10 neurons and $N = 3$–5. One-way analysis of variance (ANOVA) Bonferroni's test was applied and *p* values were indicated on every figure. The standard error of the means (SEMs) were calculated and are represented in the error bars of the plots.

In the live cell imaging, statistical tests were applied using one-way ANOVA Bonferroni's test. Three independent experiments were performed on 13 neurons for $H_2O_2$, 10 neurons for DTT, 11 neurons for NMDA and 11 neurons for NMDA in the presence of APV. The total number of spines analysed were as follows: 125 spines for $H_2O_2$, 71 spines for DTT, 94 spines for NMDA and 88 spines for NMDA in the presence of APV. To establish specificity, the mean values of NMDA-treated spines were normalized to the mean value of control spines (addition of medium only). Three COS cells were analysed for $H_2O_2$ and DTT.

For the spine density analysis, we applied one-way ANOVA Bonferroni test. On average, over 1.000 spines were detected per neuron avoiding regional bias. *N* represents the number of independent experiments and *n* the number of single neurons. For each genotype, rescue and treatment $n \geq 30$ neurons and $N = 3$. For the spine density analysis of ATM neurons, $N = 3$ and $n = 20$–30 neurons.

To measure the strength of the linear relationship between pS647 and DBN in the immunohistochemistry, we used the Spearman's correlation coefficient.

Western blot analysis was performed in three independent experiments using one-way ANOVA Bonferroni's test (multiple conditions) or *T* test (two conditions).

For the lifespan analysis, the log-rank test (Mantel–Cox) was carried out to compare the survival distributions between different groups in the lifespan assays.

In the progeny assay and fluorescence intensity measurements, Student's *t* test and one-way ANOVA Bonferroni test were carried out to test significance between different nematode strains/differently treated groups.

**Reporting Summary.** Further information on experimental design is available in the Nature Research Reporting Summary linked to this article.

## Data availability
Data generated or analysed during this study are included in this published article (and its Supplementary Information Files). All other relevant data are available from the authors upon reasonable request.

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

## Acknowledgements

We thank Kerstin Schlawe, Kristin Lehmann and Beate Diemar for excellent technical assistance. We would like to thank the viral core facility for the production of the viruses, the peptide synthesis core facility for the production of the Aβ peptide, the Advanced Medical BioImaging Core Facility (AMBIO) for providing the live cell spinning disk system and the Neurocure Multi-user Microscopy Core Facility for the usage of the confocal. We also thank Frédéric Ebstein for the help with the ubiquitination assays, Ina Bartnik for help with FUNCAT-PLA, Elke Krüger for the HA-ubiquitin-GFP construct, QueeLim Ch'ng for helpful discussions, Clemens Schmitt for kindly providing us with the ATM−/+ mice and Dieter Klopfenstein for the anti-DBN-1 *C. elegans*-specific antibody. Funding was provided by the DFG (Project 285933818, SFB958 A16), NeuroCure EXC257 + KI1988/3-1.

## Author contributions

P.K., C.G., E.R.-P., T.G.A.M., C.K., C.W., K.M. and S.t.-D. performed the experiments. P.K., C.G., E.R.-P., T.G.A.M. and C.K. analysed the experiments. V.D. developed the automated program for the PLA quantification. S.t.-D. and E.M.S. supervised the PLA experiments. P.K., T.G.A.M., J.K. and B.J.E. designed the study. P.K., C.G., T.G.A.M., J.K. and B.J.E. wrote the manuscript.

## Additional information

**Competing interests:** The authors declare no competing interests.

