## [Peer Review File · Nature Communications]

Reviewers' comments:

Reviewer #1 (Remarks to the Author):

Kreis et al demonstrate a role for the Drebin (DBN) protein in actin filament stability in dendritic spines, particularly under stress conditions. Phosphorylation of S647 in DBN is shown to regulate DBN stability. Phosphorylation of DBN is observed to increase with stress, although there is considerable basal phosphorylation. It is proposed that the ATM protein kinase phosphorylates this site under stress conditions, and purified ATM is shown to phosphorylate DBN in vitro, in agreement with this hypothesis. Lack of stress-induced DBN phosphorylation is demonstrated in ATM-deficient neurons. Lastly, the model is tested in *C. elegans* where ATM status is linked to lifespan. Mutation of DBN to S647A is shown to reduce the lifespan of *C. elegans* and also to block the effects of ATM inhibitor on lifespan.

This is an interesting paper that demonstrates that the phosphorylation of this actin-binding protein definitely affects its turnover, and that this modification is sensitive to ATM kinase modulation. There is not much functional analysis in the rodent neuron models though, and ATM-deficient mice do not have a neurological phenotype. The authors also show that DBN affects total lifespan in *C. elegans* though, and the modification at S647 seems to be important for the role of ATM. This is compelling, but it does not seem to be clear what DBN is doing to promote lifespan.

Overall, the authors provide convincing evidence showing that ATM dependent phosphorylation on DBN leads its stabilization after oxidative stress and that this event is correlated with lifespan extension. The mechanisms underlying this are not known but this may help to answer fundamental questions related to the A-T phenotype including progressive neuronal degeneration.

Specific points that should be addressed:

1. Fig 2a: Phosphorylated DBN should be normalized with total DBN also.
2. Page 8: Authors mention "ATM does not control steady-state DBN phosphorylation (Fig. 4h)". Is this correct? Or was other phosphorylation non-specifically recognized by phospho-specific antibody since there was very weak phosphorylation signal with KU55933 when recombinant DBN was incubated with brain lysate in Fig 4d. To confirm this, cells expressing S647A mutant should be used.
3. 4e/f: what is being shown in the image in 4e - what is the excitation wavelength used here? Images from excitation at both 405 and 488 nm should be shown. Labeling of Fig. 4f could also be more clear by labeling "before treatment" and "after treatment". What is being shown in the image here - presumably this is emission measured at 509 nm but the excitation wavelength is varied? This needs to be much better explained in the legend and also it would be helpful to show separate images at the optimal excitation wavelengths for each state for this to be understandable.
4. Fig 4i: Are the samples from ATM+/+ and ATM-/- loaded in the same gel? If so, Total DBN protein levels are quantitated by normalization with tubulin. ATM-/- cells should contain statistically less total DBN since authors suggest that phosphorylation of DBN induce its stabilization.
5. Related to point #4: 4g: the blot of total ATM absolutely needs to be shown here; another protein like tubulin should not be the blotting control.
6. Fig 4i: Authors should label right two panels which show quantitated pS647-DBN/Tubulin with ATM+/+ and ATM -/-.

7. Fig 4k: It would be better to check spine density in ATM^{-/-} cells expressing wt, S647A, or S647D DBN. If phosphorylation at serine 647 on DBN is the only factor which is related to spine density, S647D expression in ATM^{-/-} should be higher spine density than other.

8. Fig. 4: since ATM seems not to affect basal levels of S647 phosphorylation but alters the stress-induced phosphorylation, can the authors determine where in the cell this additional phosphorylation occurs by doing IF with phospho-647 antibody? Also, this could be tested in combination with phospho-ATM or total ATM antibody to provide evidence for co-localization of the proteins in the neurons. Even though ATM purified from cells does phosphorylate this site in vitro, it is not as clear that this is a direct phosphorylation event in cells since many different kinases could be downstream of ATM.

9. Fig. 5d: too many curves on top of each other.

Reviewer #2 (Remarks to the Author):

This study by Kreis et al utilizes a plethora of different techniques, appears to be conducted rigorously, and provides exciting conclusions that are well supported by the data. The manuscript is also generally very well written, in a succinct yet clear manner. While there are several points that should be addressed, none of them require additional experiments. Overall, this study seems highly suitable for publication in Nature Communications.

Major points:

1) Title: I understand why the authors wish to include "lifespan" in the title. However, I believe this is misleading for two reasons: (i) The data do not show a control of lifespan itself, but a sensitization to the cumulative toxic effect of continuously increased oxidative stress; and (ii) the corresponding experiments were done in worm, and thus require this qualifier (as the other data collected largely concern dendritic spines in cultured mammalian neurons).

2) Results Fig. 1e: Contrary to the statement in the text, 1 μ M Abeta is not a low dose. Even 0.5 μ M Abeta have been consistently reported to cause a robust decrease in spine number. Only reducing concentration to 0.1 μ M (which is still sufficient to robustly inhibit LTP) should start having no effect in WT neurons. This requires at least some discussion (and repetition in multiple independent culture preparations). One reason could be differences in the preparation of the soluble Abeta oligomer (i.e. typically the starting concentration is well defined, but the amount of actually obtained soluble oligomers may not be). Nonetheless, rather than stating that the concentration is "low", it needs to be specifically stated that typically, this concentration reduces spine density in wild type. Also, the time of incubation with Abeta does not appear to be stated anywhere in the manuscript; this has to be included.

Minor point:

3) Results Fig. 1g: dbn ko has no effect on spine density, but re-expression increases it. This needs to be discussed a little bit, especially since it slightly complicates the interpretation of the similar (though slightly larger) difference in presence of paraquat. The conclusion still appears valid, but this complication needs to be at least mentioned.

Extremely minor comments:

4) First reference to Fig. 1a is not quite accurate; could be alleviated by referring to the whole Fig. 1 after the first half of the sentence.

5) The first text description of Fig. 2a should mention development; the phrase "dynamic pattern" is not helpful and if any unnecessarily misleading. Maybe actually mention the specific result.

6) Description of Fig. 4f could mention that the NMDA effect was sensitive to the NMDAR inhibitor APV.

7) Description of Fig. 4i should describe the experiment, i.e. how neuronal activity was induced.

8) The Western in Fig. 5a is not mentioned in the text, and the description in the Fig legend is somewhat unclear. Is the antibody specific to human dbn (i.e. doesn't detect worm dbn)? Why are there two bands? And why is there no lane for WT worm (and dbn ko worm)?

9) Legend Fig.1a: CaMKII localizes in both spines and dendrites (not just spines), as expected.

10) Legend Fig. 1e: why "globular"? Also, Abeta concentration and incubation time should be mentioned.

11) Fig 1f: At 0.1 uM paraquat, is the reduction in WT spine density significant? Fig. 1g: 1st and 4th column difference significant? i.e. maybe indicate "n.s." as in panel d and e?

12) Fig 1g, right bar graph: change y-axis label (to something like "spine phalloidin intensity", or "spine F-actin content").

13) Statistics: for each experimental analysis is indicated in each methods subsection. It would be more useful to do this instead either in each Figure legend, or -alternatively- in a separate statistics section in the methods (makes it easier to find, and avoids repetition)

14) Fig. 4a: add some lines in order to indicate localization of the shown sequence in the stick diagram.

Reviewer #3 (Remarks to the Author):

In this manuscript Kreis et al. provide evidence that phosphorylation of the actin filament binding protein drebrin at S647 by the kinase ATM controls oxidation stress resistance and lifespan. At first, they showed that the dendritic spines on cultured hippocampal neurons from DBN^{-/-} mice are more sensitive to the actin filament depolymerizing compound latrunculin B and the synaptotoxic effects of Aβ oligomers than wild type neurons. These results prompted them to investigate oxidative stress and they showed that cultured hippocampal neurons from DBN^{-/-} mice are more sensitive to paraquat than wild type. They then directed their attention to studying pS647 drebrin

arguing that drebrin phosphorylation might control drebrin turnover and showed that a S647 phospho-dead mutant of DBN has a shorter half-life in HEK 293T cells than a phospho-mimetic mutant or wild type DBN. They identified a kinase (ATM) that can phosphorylate drebrin at S647 in vitro and in a cell and showed that oxidative stress and increased neuronal activity increases ATM phosphorylation of drebrin, which is not seen in the ATM^{-/-} mouse. Interestingly, drebrin in the ATM^{-/-} mouse is still phosphorylated at S647 implying the existence of a separate, constitutively active S647 kinase. Importantly, paraquat induced loss of dendritic spines was exacerbated in ATM^{-/-} neurons and WT and DBNS647D but not DBNS647A rescued this effect. Finally, in experiments with *C. elegans* mutants the authors show lifespan increases and resistance to stress (paraquat treatment) depends on DBN S647 phosphorylation and ATM.

I enjoyed reading this paper and, in general, the experiments are clearly described, the data is of good quality and the conclusions are, overall, justified by the data. The major findings are novel and will be of general interest and stimulate thinking in the field.

1. The claim that LatB did not affect dendritic spine morphology (line 63) is not substantiated since this was not measured. Here and elsewhere in the manuscript dendritic spine density is measured but not morphology. I think that this is a missed opportunity for insight into mechanism. Measuring overall spine loss is a crude measure of spine dynamics and hides potential loss of or morphological changes, such as shrinkage, in particular spine types, including filopodia.
2. The authors conclude from the loss of dendritic spine density that there is an effect of DBN levels on “net actin filament stability” (line 69). However, this has not really been measured directly but is mainly inferred from the literature. Other mechanisms might be involved, for example DBN has been shown to enhance microtubule insertion into dendritic spines which increases spine stability (Merriam et al., *J. Neurosci.*, 33, 16471-82, 2013). While the latrunculin experiment is consistent with an effect on actin filament stability the converse experiment has not been done. Stabilising actin filaments with jasplakinolide, for example, should antagonise the effects of stress. This has been done recently by the Shirao lab (Hanamura et al., *Neurosci.*, 379, 67-76, 2018).
3. A distinction is not made between the embryonic form of drebrin and the adult form. The adult form (drebrin A) has a ~45 amino acid long insert of unknown function down-stream of the coiled-coil region. It is not clear in all instances which form is being used experimentally. Is DBN 1, transcript variant 1 (line 330), drebrin E and DBN1 iso3 (line 339), drebrin A? Also, in Fig. 2a there is no mention of the fact that the pS142 site is only present on the embryonic form (drebrin E) whereas pS647 is present on both, i.e. the upper and lower bands in the immunoblot.
4. Immunoblots should be quantified when statements are made about protein levels (e.g. Figs 2f and 4c). And in the case of phospho-DBN levels they should be compared with DBN total and not normalised to tubulin as in Fig. 4i right-hand panel.
5. The discovery of a “steady-state” kinase that phosphorylates S647-DBN in ATM^{-/-} mutant mouse neurons is very interesting. How is the level of phosphorylated drebrin by this unknown kinase regulated? Since ATM phosphorylates DBN directly, rather than say by inhibiting a S647-DBN phosphatase, there must be a reserve pool of unphosphorylated DBN that ATM phosphorylates. It should be possible to see this pool biochemically by quantitative immunoblotting and it would be interesting to know where in the neuron this pool is. Examining the location of pS647 DBN in ATM^{-/-} mutant mouse neurons might shed some light on this question. Also, since this group has shown that PTEN de-phosphorylates pS647-DBN (reference 19) have they ruled out whether ATM can inhibit PTEN?

Line 59, "cellular insult or stress renders spines vulnerable to DBN-loss", surely the other way around?

Line 122, HEK 293T

Line 160, absence and or presence

Line 247, why is Fig. S5 before Fig. S4? (line 254)

Line 309, 30 ng

Line 471, were PTEN antibodies used?

Line 754, n.s. not used in figure

Figure 5a, why are there two bands for drebrin in the immunoblot and which one is hDBN-YFP? Probably the 130 kDa band as judged by the blot in Fig S3 (N2), although there is also a band at 100 kDa in this blot. The 180 kDa band can't be dbn-1 because this is a dbn

Reviewer #4 (Remarks to the Author):

Kreis et al. examines the role of DBN (developmentally regulated brain protein), a synaptic protein that, like many, showed promise to neurobiologists first by locating at the dendritic spines, showing activation-dependent phosphorylation, dynamic binding to PTEN, etc.), then disappointing them with a knockout rodent that exhibits no gross developmental, functional, and, behavioral (?) defects.

Studies from these authors suggest that DBN is likely important after all, under oxidative stressed condition, and during related physiological conditions such as aging. They show following key results:

1) In isolated primary hippocampal neuron cultures, the loss of DBN rendered them more sensitive to dendritic spine loss induced by oxidative or A β induced stress, and the protective role of DBN depends on DBN's ability to phosphorylate at aa647.

2) Through biochemistry, pulse-chase, and imaging experiments, using cultured hippocampal primary neurons, or HEK cells, they demonstrated that the DBN was likely locally synthesized at dendrites, and the phosphorylation at 647aa regulates its UPS-mediated protein turnover. But the most informative mechanistic study is their demonstration that when subjected to oxidative stress, a PI3-like, DSB- and ROS-activated kinase, ATM, enhances DBN's phosphorylation at aa647. Disruption of ATM reduces DBN's activity-dependent modulation of s647, and DBN's protective function.

3) The relation between oxidative stress and aging is better addressed in *C. elegans* in a whole organism level. For in vivo studies, they examined the role of mammalian DBN using the *C. elegans* as a host. By expressing DBN, DBN-S65A, and DBN-S65D in *C. elegans* *dbn-1* mutants, they determined that the phosphorylation status of DBN regulates aging under oxidative stress, and such an effect may also be dependent on ATM, because the effect was partially blocked by an ATM inhibitor.

4) Results from in vitro and in vivo studies using different experimental systems implicate that ATM-mediated activity-dependent s65 DBN phosphorylation may influence actin dynamics and dendrite maintenance under oxidized or metabolic stressed conditions.

Overall assessment: This group has been studying the role of DBN, a synaptic protein with little knowledge on its physiological function. Establishing a mechanistic link between DBN and ATM is a novel finding and key breakthrough. Placing the role of DBN in stress and aging, in part through ATM, and vice versa, is of significant biological significance.

This study utilized a wide-array of experimental systems (primary hippocampal neuron cultures, hippocampal slices, rat brain slices and cortical neuron cultures, and *C. elegans*), and advanced experimental techniques, from biochemistry (pulse-chase and ubiquitination assay) to advanced cell imaging (FUNCAT-PLA and FRET) in cultured mammalian neurons and cells, as well as CRISPR in *C. elegans*, to address the role of DBN and its functional relationship with ATM. Experiments were well designed and performed with rigor. Conclusions are convincing, and it was well written, making it pleasant to read through.

I support its acceptance for publication, upon authors addressing a few minor concerns and suggestions.

1. In Fig. 2C's pulse chase experiments, authors used HEK293T cells and transfected DBNs to determine the effect of s65 phosphorylation on its half-life. In Fig. 4, the effect of oxidation-induced ATM activation for DBN phosphorylation was examined in HEK293T cells that express ATM and DBN endogenously (Page 7, line 167). It was clear (from methods or the text) whether there are two different HEK293T cell lines that exhibit difference in the endogenous level of DBN and ATM, or they are really just the same line, and authors examined the endogenous DBN protein in the second experiment. If this is the case, it may be more relevant to determine the half-life of protein at the endogenous level, although it does make comparison of the half-life of DBN S65D or DBN S65A difficult.

2. Using ATM^{-/-} mice, as well as the ATM inhibitor, authors convincingly demonstrated a requirement of ATM for stress- or activity-dependent increase of S65 DBN phosphorylation. Is there any expression data that corroborate the either steady-state or activity-dependent localization of ATM to dendrites? In addition, steady-state phosphorylation at s65-DBN seems to be dependent on other, unidentified kinases. Authors should make this clear to readers.

3. Comparatively, the *C. elegans* studies are not solid or as well presented, and it needs a bit more work.

- I am impressed that in addition to order a deletion strain from the stock center that does not lead to a complete KO (ok925), authors generated a true knockout of the *C. elegans* homologue by CRISPR (JKM1) in order to have true complete KO. But authors never presented data to compare the phenotype of the two strains, or the hinted successful removal of residual function of DBN by feeding RNAi in the ok925 background (Methods; Page 21). This is critical because authors used different genetic background to generate the 'humanized' DBN strains. They expressed DBNS65A or D proteins in ok925, whereas DBN wild-type in the JKM1 background (Methods; Page 1). To compare their effect on aging and oxidative stress fairly, authors should make sure that the background (*C. elegans* DBN^{-/-}) should be equal, and the expression level of the three integrated transgenes to be at least similar. There should be quantified data for this.

- Authors did not show whether the *elegans* dbn mutants have an aging-related phenotype when compared to wild-type animals. This is critical. The conclusion of these studies should suggest that dbn mutants exhibited reduced aging, or more sensitivity to PA-induced aging. Overall, this part of the study reads like an addition that ignores the rich biology of an organism, and reduces it simply to be a test-tube to compare the effect of overexpressing vertebrate DBNs by a convenient assay. Though this may be a trend of thinking of the neurodegenerative field, it is still a bit painfully wasteful way to use this system for aging, because it has so much more to offer for true mechanistic discoveries when used properly.

- Similar to the above, the dependence of the DBN phenotypes on ATM was examined by the ATM inhibitor, which was only partially effective. Caveats remain regarding its specificity or efficacy to the endogenous *C. elegans* ATM. Authors could at least discuss this caveats, and better approaches, such as using atm ko mutants, for future studies if they do want to address this process using the *C. elegans* experimental system.

4. Overall the manuscript is prepared in such compact format, while very well prepared, had little room to discuss backgrounds and caveats, and so much experimental details that are critical for the interpretation of the results, such as the specific cell culture type, their different DIV stage, different genetic backgrounds, were all left at the Methods for readers to find. Authors should reformat the manuscript for Nature Communications so that proper results sections can be read to help both the readers understand and appreciate the rigor that went into each set of their experiments.

Just as another example to illustrate this point: In both Fig. 5 and Fig. S5, DBN antibodies revealed two bands, 180Kd and 130Kd, respectively. Are they two isoforms of DBN that both could be phosphorylated at S65? It seems so but there was no description or explanation of such information that I could easily find. If this is the case though, it was not explained in all other western blot, which only the 130kd isoform was examined in all biochemistry experiments shown in the paper.

We thank the four reviewers for their overall very positive and encouraging comments. They note the importance and novelty of the work in demonstrating that, for example, 'phosphorylation of this actin-binding protein definitely affects its turnover, and that this modification is sensitive to ATM kinase modulation' (Reviewer 1). Reviewer 2 refers to the 'plethora of different techniques' and that the work 'appears to be conducted rigorously and provides exciting conclusions that are well supported by the data'. We are particularly happy to find that this reviewer found that 'the manuscript is also generally very well written, in a succinct yet clear manner', which was also reflected in the comments of reviewer 3, who 'enjoyed reading this paper'. Finally, reviewer 4 remarked on the fact that 'establishing a mechanistic link between DBN and ATM is a novel finding and key breakthrough. Placing the role of DBN in stress and aging, in part through ATM, and vice versa, is of significant biological significance'.

In this light, we have now addressed most of the reviewers concerns:

- We have extended the work regarding the life span experiments and undertaken further work to strengthen the claims on lifespan and aging-related phenotypes in *C. elegans*.
- We have provided additional quantifications of western blots
- We have included a completely new set of experiments that test the function of DBN phosphorylation in neurons obtained from ATM^{-/-} mice during oxidation-induced stress.
- We have tested a number of different protocols with a view to characterizing the localization of endogenous ATM. However, we believe ATM antibodies routinely used in biochemical applications are not convincing as tools in immunocytochemical applications, due to the lack of success in obtaining significant decreases in ATM or pATM signals in neurons obtained from ATM^{-/-} mice. We have discussed this issue at length in the rebuttal letter below.
- In adherence to the recommended editorial policy checklist, we have replaced the red color in the RGB color-coding of our images by magenta and have included the dot plots into the bar graphs.

Response to Reviewer's Comments

Reviewer #1 (Remarks to the Author):

1. Fig 2a: Phosphorylated DBN should be normalized with total DBN also.

• *We performed two additional experiments using rat brain P7 and W20 protein lysates, which are the developmental stages that demonstrate the most significant changes in the time course demonstrated in Figure 2a. Quantifications of pS647-DBN normalized to total DBN and pS647-DBN normalized to tubulin from 3 independent experiments are now represented in the histogram (Fig 2a, right). As expected, although protein levels of pS647-DBN normalized to tubulin decrease significantly between P7 and W20, levels of pS647-DBN normalized to total DBN remain constant, indicating that pS647-DBN levels follow precisely the pattern of expression of total DBN.*

2. Page 8: Authors mention "ATM does not control steady-state DBN phosphorylation (Fig. 4h)". Is this correct? Or was other phosphorylation non-specifically recognized by phospho-specific antibody, since there was very weak phosphorylation signal with KU55933 when recombinant DBN was incubated with brain lysate in Fig 4d. To confirm this, cells expressing S647A mutant should be used.

• *The pS647-DBN antibody has been characterized in depth in Kreis, P. et al. 2013 (Phosphorylation of the actin binding protein Drebrin at S647 is regulated by neuronal activity and PTEN. PLoS One 8). In this paper, we demonstrate that the pS647-DBN antibody does not recognize*

YFP-DBN-S647A expressed in HEK cells. Furthermore, we also show that knocking down DBN using shRNA reduced pS647-DBN detection in the same way the pan anti-DBN antibody does. Therefore, we demonstrate that the pS647-DBN is a very specific antibody for the S647-phosphorylated form of DBN. (see <https://journals.plos.org/plosone/article?id=10.1371/journal.pone.0071957>)

- Concerning Fig. 4d, we believe that the weak band detected with anti-pS647-DBN in the presence of the ATM inhibitor (KU55933) is due to the phosphorylation of DBN by other kinases present in the brain lysate. Indeed, we show in Fig. 4h that in the absence of ATM, Drebrin retains some phosphorylation at S647 indicating that ATM kinase is not the only kinase responsible for DBN phosphorylation at S647.

3. Figure 4e/f: what is being shown in the image in 4e - what is the excitation wavelength used here? Images from excitation at both 405 and 488 nm should be shown. Labeling of Fig. 4f could also be more clear by labeling "before treatment" and "after treatment". What is being shown in the image here - presumably this is emission measured at 509 nm but the excitation wavelength is varied? This needs to be much better explained in the legend and also it would be helpful to show separate images at the optimal excitation wavelengths for each state for this to be understandable.

- We agree that this part of this figure could have been better presented. We have now made changes that, we hope, will improve the presentation and explanation of ro-DBN. Fig. 4e and Fig. 4f now includes both images at excitation 405 nm and 488 nm. The Figure legend has been further detailed and the labeling of Fig. 4f has been changed to (-) before and (+) after the indicated treatment.

4. Fig 4i: Are the samples from ATM+/+ and ATM-/- loaded in the same gel? If so, Total DBN protein levels are quantitated by normalization with tubulin. ATM-/- cells should contain statistically less total DBN since authors suggest that phosphorylation of DBN induce its stabilization.

- We thank the reviewer for this comment. The western blot shows the same exposure of all samples that were loaded, and separated on one gel; however, we decided to crop the gel because of the way in which results are presented in all other figures throughout the paper (i.e. first untreated and then treated conditions). We believe that this presentation is more consistent with the overall study and will help the reader to evaluate the data. We have included the entire (uncropped) western blot below for inspection below. In Figure 4i, we have now included the quantification of DBN/Tubulin. As required by reviewer 3 in point 4 (see below), we have also changed the quantification of pS647-DBN/tubulin to pS647-DBN/DBN.

5. Related to point #4: 4g: the blot of total ATM absolutely needs to be shown here; another protein like tubulin should not be the blotting control.

- *Again, we thank the reviewer for requesting those changes. Three new experiments have been performed and the relative p-ATM/ATM levels have been quantified. We believe that this inclusion unequivocally demonstrates an increase in p-ATM over total ATM levels.*

6. Fig 4i: Authors should label right two panels which show quantitated pS647-DBN/Tubulin with ATM^{+/+} and ATM^{-/-}.

- *Labelling of the figures has been modified according to the reviewer's suggestion. The quantification has also been changed to pS647-DBN/DBN as requested by reviewer 3 in point 4.*

7. Fig 4k: It would be better to check spine density in ATM^{-/-} cells expressing wt, S647A, or S647D DBN. If phosphorylation at serine 647 on DBN is the only factor which is related to spine density, S467D expression in ATM^{-/-} should be higher spine density than other.

- *We thank the reviewer for this excellent suggestion and have performed a whole new set of experiments that test the hypothesis posed by the reviewer. We performed expression of the DBN-S647A mutant, as well as the DBN-647D mutant in hippocampal neurons of ATM^{-/-} mice and measured spine density under the influence of oxidative stress as before. Expression of DBN-S647D, but not DBN-S647A, is able to protect ATM^{-/-} neurons from oxidative stress. We included the new data in the manuscript (Fig 5c).*

8. Fig. 4: since ATM seems not to affect basal levels of S647 phosphorylation but alters the stress-induced phosphorylation, can the authors determine where in the cell this additional phosphorylation occurs by doing IF with phospho-647 antibody? Also, this could be tested in combination with phospho-ATM or total ATM antibody to provide evidence for co-localization of the proteins in the neurons. Even though ATM purified from cells does phosphorylate this site in vitro, it is not as clear that this is a direct phosphorylation event in cells since many different kinases could be downstream of ATM.

- *pS647-DBN and DBN are enriched in dendritic spines of mature hippocampal neurons (Kreis et al., 2013; figure below).*

- *To label ATM we tested two different anti-pan ATM antibodies by western blot and by immunofluorescence that are used routinely by a number of labs. In western blot applications, we detected consistently a band at around 350 kDa in ATM^{+/+} neurons but not in the ATM^{-/-} neurons (below, arrow). However, many additional and intense bands were detected using both antibodies, even after testing different protocols for antibody detection of protein on membranes.*

- We used previously described antibodies and protocols for ATM detection by immunofluorescence in neurons, see for example Li et al., *Cytoplasmic ATM in neurons modulates synaptic function* *Curr. Biol.* 19, 2091–2096, 2009; or Li et al., *Stable brain ATM message and residual kinase-active ATM protein in ataxia-telangiectasia* *J. Neurosci.* 31, 7568–7577, 2011; and obtained very encouraging synaptic localizations of ATM that were also clearly enriched in dendritic spine compartments. However, we tested the antibodies additionally in ATM^{-/-} neurons, which resulted consistently in an equally intense fluorescent signal (see below example of anti-ATM 2C1(1A1)).

- In the previous papers, the authors never used ATM^{-/-} mice in parallel to test unequivocally for specificity of the obtained signals. We also tested further a number of fixation protocols and labelling conditions, but in all cases, we were never 100% convinced that signals, even the most compelling signals in dendritic spines, are indeed specific. We have inserted an additional figure below that demonstrates co-localization of anti-ATM labeling and pS647-DBN labelling in the spine compartment. Nevertheless, we would much rather not include this result due to the reasons detailed above.

Given that DBN resides exclusively in the post-synaptic compartment, and that ATM unequivocally phosphorylates DBN, it is highly probable that ATM and DBN colocalise at the dendritic spine. Despite the fact that we cannot show the colocalisation directly, it still seems a valid conclusion.

- We also tested pS1981-ATM antibody by further western blot experiments, as well as by immunofluorescence. The new biochemical analyses of pATM/ATM have been included in Figure 4g.

The labelling of neurons by pATM was, however, not satisfactory. In all conditions tested, the fluorescence intensity of anti-pATM immunoreactivity in wt neurons remained similar to the anti-pATM signals detected in ATM^{-/-} neurons. Similarly, using an immunocytochemistry approach, we were unable to detect changes in pATM-signals in response to biccuculine treatment in neurons.

- We also embarked on the analysis of ATM localization at the synapse by preparing synaptosomes from whole adult brain. We were able to detect ATM in the crude synaptome (P2) as well as in the cytoplasm (S2). Whilst this method indicates the presence of ATM at the synapse, it cannot be applied to the task of distinguishing between pre- and post-synapse compartment.

9. Fig. 5d: too many curves on top of each other.

The comment that figure 5d is too crowded is shared by all reviewers and we have therefore rearranged this figure. We split the lifespan curves into different sub-figures to allow a comparison of a limited set of variables in each sub figure.

The previous figure 5d depicted the lifespan data of:

$\Delta dbn-1/nDBN^{S647A}$

$\Delta dbn-1/DBN^{S647D}$

$\Delta dbn-1/nDBN^{S647A}$ + ATM inhibitor

$\Delta dbn-1/nDBN^{S647D}$ + ATM inhibitor

And all 4 conditions with and without paraquat treatment.

We have now separated the data sets and show for better clarity:

figure 6b) $\Delta dbn-1$ and $\Delta dbn-1/nDBN^{wt}$ (previously depicted in figure 5c)

figure 6c) $\Delta dbn-1$ and $\Delta dbn-1/nDBN^{wt}$ +/- ATM inhibitor

figure 6d) $\Delta dbn-1/nDBN^{S647A}$ and $\Delta dbn-1/nDBN^{S647A}$ +/- ATM inhibitor (depicted in red and magenta); $\Delta dbn-1/DBN^{S647D}$ and $\Delta dbn-1/nDBN^{S647D}$ +/- ATM inhibitor (depicted in blue and dark blue)

All conditions with (dashed line) and without (solid line) paraquat treatment.

The table summarizing the median half-live data of all tested conditions is now depicted in figure 6e (5e in previous version) and figure S3e (for N2, RB1004 and JKM1 +/- paraquat and ATM inhibitor).

We agree with the criticism made by all of the reviewers on this point. We hope that the new layout adopted will make it easier to understand what remains a highly complex dataset.

Reviewer #2 (Remarks to the Author):

Major points:

1) Title: I understand why the authors wish to include "lifespan" in the title. However, I believe this is misleading for two reasons: (i) The data do not show a control of lifespan itself, but a sensitization to the cumulative toxic effect of continuously increased oxidative stress; and (ii) the corresponding

experiments were done in worm, and thus require this qualifier (as the other data collected largely concern dendritic spines in cultured mammalian neurons).

We believe that the inclusion of 'lifespan' in the title is justified, as it is a standard readout using in the scientific community of researchers using the C. elegans model. We believe that by demonstrating an effect of DBN phosphorylation during oxidation induced stress, our results can be referred to an experimental setting that tests life-span.

2) Results Fig. 1e: Contrary to the statement in the text, 1 μ M Abeta is not a low dose. Even 0.5 μ M Abeta have been consistently reported to cause a robust decrease in spine number. Only reducing concentration to 0.1 μ M (which is still sufficient to robustly inhibit LTP) should start having no effect in WT neurons. This requires at least some discussion (and repetition in multiple independent culture preparations). One reason could be differences in the preparation of the soluble Abeta oligomer (i.e. typically the starting concentration is well defined, but the amount of actually obtained soluble oligomers may not be). Nonetheless, rather than stating that the concentration is "low", it needs to be specifically stated that typically, this concentration reduces spine density in wild type. Also, the time of incubation with Abeta does not appear to be stated anywhere in the manuscript; this has to be included.

- *We thank the reviewer for raising this issue. Prior to testing $A\beta_{1-42}$ on neurons isolated from DBN^{+/-} and DBN^{-/-} neurons, we tested neurons with our $A\beta_{1-42}$ oligomer preparation on WT neurons and determined the concentration, in which oligomers do not decrease spine density in 3 independent neuronal culture preparations. The reviewer is correct that stating that the term "low" is not appropriate here and we have removed it from the text. Several studies have shown dendritic spine loss in the presence of 1 μ M or lower oligomeric $A\beta$. However, the experimental conditions in these studies often differ from our setup and this could explain the differences observed. One example is the article from the Kinney lab (Neutralization of soluble, synaptotoxic amyloid β species by antibodies is epitope specific, Zago et al., 2012), where dendritic spine density was analyzed using the dendritic marker Spinophilin in rat hippocampal neurons. Here dendritic spine density decreases with 500nM $A\beta_{1-42}$ oligomer. As suggested by the reviewer, we have inserted in the text a statement that typically 1 μ M $A\beta_{1-42}$ oligomer reduces spine density in wild-type rat neurons (see page 3, lines 47-49).*

- *The time of incubation is indicated in the legend of fig. 1e.*

Minor point:

3) Results Fig. 1g: dbn ko has no effect on spine density, but re-expression increases it. This needs to be discussed a little bit, especially since it slightly complicates the interpretation of the similar (though slightly larger) difference in presence of paraquat. The conclusion still appears valid, but this complication needs to be at least mentioned.

- *The re-expression of DBN-YFP in dbn KO neurons does indeed increase spine density in the absence of paraquat. We believe that neurons in culture are susceptible to stress by the in vitro culture itself and this may explain the protective effect of Drebrin re-expression. We have now included a statement to this effect in the text (page 4, line 59).*

Extremely minor comments:

4) First reference to Fig. 1a is not quite accurate; could be alleviated by referring to the whole Fig. 1 after the first half of the sentence.

- *This has been modified accordingly*

5) The first text description of Fig. 2a should mention development; the phrase "dynamic pattern" is not helpful and if any unnecessarily misleading. Maybe actually mention the specific result.

- *Dynamic pattern has now been changed to developmental pattern in the text.*

6) Description of Fig. 4f could mention that the NMDA effect was sensitive to the NMDAR inhibitor APV.

- *A sentence describing the antagonistic effect of the NMDA receptor inhibitor APV has been included in the main text (page 8, line 173).*

7) Description of Fig. 4i should describe the experiment, i.e. how neuronal activity was induced.

- *We have inserted in the main text a description of the fact that neuronal activity was induced using bicuculline, a GABA-A receptor antagonist (page 8, line 189).*

8) The Western in Fig. 5a is not mentioned in the text, and the description in the Fig legend is somewhat unclear. Is the antibody specific to human dbn (i.e. doesn't detect worm dbn)? Why are there two bands? And why is there no lane for WT worm (and dbn ko worm)?

We thank the reviewer for raising these two valid points: On the one hand, he/she refers to the presence of an additional band detected by the DBN antibody. On the other hand, he/she points out that we should present expression of the nematode DBN-1 in the wild type and upon depletion.

- *To address the first point, we have optimized the lysis protocol to avoid any processing of the human DBN-YFP. The additional band in the previous manuscript resulted from proteolysis. We have now included further protease inhibitors (see Materials and Methods) to avoid any degradation during lysis and processing of the samples prior to SDS-PAGE and western blot analysis. Figure S3g depicts the human DBN variants expressed in the nematode. We used antibody M2F6 to detect human Drebrin. We also used the same new protocol to lyse nematodes treated to KU55933 and detected a single band for human DBN phosphorylation. The new western blot is depicted in figure S4c.*

- *To address the second point, we have analyzed the expression of the truncated nematode DBN-1 in strain RB1004 and the humanized Drebrin strains +/- dbn-1 RNAi treatment to deplete the truncated DBN-1 of the nematode and show the western blot in figure S4a. We used the C. elegans-specific DBN-1 antibody provided by the Klopfenstein lab (Butkevich et al., 2015) to detect the nematode DBN-1. A western blot of the CRISPR/Cas generated complete knockout of dbn-1 is shown in figure S4b.*

We hope that we have provided sufficient new data to address this reviewer's concerns.

9) Legend Fig.1a: CaMKII localizes in both spines and dendrites (not just spines), as expected.

- *We thank the reviewer for our oversight. We modified the text accordingly.*

10) Legend Fig. 1e: why "globular"? Also, Abeta concentration and incubation time should be mentioned.

- *We have changed globular to synaptotoxic A β species.*
- *The A β concentration and incubation time have now been indicated in the legend.*

11) Fig 1f: At 0.1 uM paraquat, is the reduction in WT spine density significant? Fig. 1g: 1st and 4th column difference significant? i.e. maybe indicate "n.s." as in panel d and e?

- *At 0,1 uM paraquat, the reduction in spine density compared to control is not significant. As suggested, we have indicated the significance between control and 0,1 uM and between control and 1 uM for the wild type neurons as well as for the knock out neurons. In addition, we have indicated the statistical significances between wild type and KO neurons at each concentration. For space reasons,*

we decided not to compare any further the remaining conditions. We hope that these alterations will address this reviewer's suggestion.

12) Fig 1g, right bar graph: change y-axis label (to something like "spine phalloidin intensity", or "spine F-actin content").

- *The y-axis label has been modified to spine head F-actin content*

13) Statistics: for each experimental analysis is indicated in each methods subsection. It would be more useful to do this instead either in each Figure legend, or -alternatively- in a separate statistics section in the methods (makes it easier to find, and avoids repetition)

- *This is a very important point, and in answer to it we have inserted a separate statistical analysis section in material and methods.*

14) Fig. 4a: add some lines in order to indicate localization of the shown sequence in the stick diagram.

- *Localization of the shown sequence has been indicated with 2 dotted lines.*

Reviewer #3 (Remarks to the Author):

1. The claim that LatB did not affect dendritic spine morphology (line 63) is not substantiated since this was not measured. Here and elsewhere in the manuscript dendritic spine density is measured but not morphology. I think that this is a missed opportunity for insight into mechanism. Measuring overall spine loss is a crude measure of spine dynamics and hides potential loss of or morphological changes, such as shrinkage, in particular spine types, including filopodia.

- *We eliminated this line from the manuscript. Our initial analysis of spine density, length, volume and maximal diameter (i.e. spine head diameter) did not reveal significant differences between DBN-WT and DBN-KO hippocampal neurons. We therefore included only spine density data in our manuscript, since any transient shift in spine morphology caused by drebrin loss apparently does not impinge on overall spine density. We have, nevertheless, started a detailed analysis of dynamic behavior of F-actin as well as potential super-structural changes caused by drebrin loss in the hope of collecting data for a subsequent manuscript.*

- *We agree with the reviewers comment that morphological changes, such as shrinkage, may be relevant dynamic stages towards spine loss during harmful conditions that uncover deficient F-actin stability in DBN-KO neurons. However, given the potential transitory nature of these changes, we consider spine density as a valid "end point"-parameter reflecting permanent loss of spines due to failure in chronic F-actin stabilization.*

2. The authors conclude from the loss of dendritic spine density that there is an effect of DBN levels on "net actin filament stability" (line 69). However, this has not really been measured directly but is mainly inferred from the literature. Other mechanisms might be involved, for example DBN has been shown to enhance microtubule insertion into dendritic spines which increases spine stability (Merriam et al., J. Neurosci., 33, 16471-82, 2013). While the latrunculin experiment is consistent with an effect on actin filament stability the converse experiment has not been done. Stabilising actin filaments with jasplakinolide, for example, should antagonise the effects of stress. This has been done recently by the Shirao lab (Hanamura et al., Neurosci., 379, 67-76, 2018).

- *We have changed "net actin filament stability" to "F-actin content". We hope that his will reflect our measurements better (page 3, line 43).*

- *The reviewer suggests that a treatment with jasplakinolide (Jasp) should antagonize the effects of stress and thus prove a direct effect of DBN on F-actin stability. Hanamura et al. (Neurosci., 379, 67-76, 2018) applied 2 μ M Jasp “to clarify whether GFP-DA (drebrin adult isoform) and GFP-DE (drebrin embryonic isoform) dynamics are influenced by actin filament stability”. They further report “consistent with previous findings, Jasp almost completely prevented FRAP of GFP-actin (data not shown)”. We started pilot experiments with 24 h treatment of oxidatively-stressed cultures using various doses of Jasp. Surprisingly, doses above 25 nM Jasp do not seem compatible with phalloidin-based detection of spines, presumably since both compete for binding sites. Moreover, treatment with 2 μ M Jasp over 24 h did not result in healthy cultures (data not shown), indicating actin treadmilling should not be blocked completely over such a long period. We therefore consider stabilizing F-actin directly to protect spines against insults to be a potentially very interesting project but anticipate that further work is required towards the development and validation of the specific assays.*

3. A distinction is not made between the embryonic form of drebrin and the adult form. The adult form (drebrin A) has a ~45 amino acid long insert of unknown function down-stream of the coiled-coil region. It is not clear in all instances which form is being used experimentally. Is DBN 1, transcript variant 1 (line 330), drebrin E and DBN1 iso3 (line 339), drebrin A? Also, in Fig. 2a there is no mention of the fact that the pS142 site is only present on the embryonic form (drebrin E) whereas pS647 is present on both, i.e. the upper and lower bands in the immunoblot.

- *Homo sapiens DBN 1, transcript variant 1(NM_004395.3) corresponds to Drebrin E and DBN1 iso3 (Q16643-3) corresponds to Drebrin A. This has been inserted into the material and methods section of the manuscript.*
- *An additional sentence has been inserted in the text on the observation that pS142 is only present in the Drebrin E isoform (page 4, line 70).*

4. Immunoblots should be quantified when statements are made about protein levels (e.g. Figs 2f and 4c). And in the case of phospho-DBN levels they should be compared with DBN total and not normalised to tubulin as in Fig. 4i right-hand panel.

- *Levels of ubiquitinated Drebrin (HA conjugates of immunoprecipitated Drebrin) related to Drebrin (immunoprecipitated DBN) have now been quantified in three independent experiments and are shown in Fig. 2f. A significant increase of ubiquitinated Drebrin was observed when the proteasome was inhibited using MG132.*
- *Levels of pS647-DBN/DBN have been quantified in three independent experiments, and are now shown in Fig.4c.*
- *The comparison of pS647-DBN levels have been changed to total DBN as shown in Fig. 4i. An additional quantification of Drebrin levels relative to Tubulin has been inserted as suggested by reviewer 1, point 4.*

5. The discovery of a “steady-state” kinase that phosphorylates S647-DBN in ATM-/- mutant mouse neurons is very interesting. How is the level of phosphorylated drebrin by this unknown kinase regulated? Since ATM phosphorylates DBN directly, rather than say by inhibiting a S647-DBN phosphatase, there must be a reserve pool of unphosphorylated DBN that ATM phosphorylates. It should be possible to see this pool biochemically by quantitative immunoblotting and it would be interesting to know where in the neuron this pool is. Examining the location of pS647 DBN in ATM-/- mutant mouse neurons might shed some light on this question. Also, since this group has shown that PTEN de-phosphorylates pS647-DBN (reference 19) have they ruled out whether ATM can inhibit PTEN?

- *Previous studies have shown that ATM can phosphorylate PTEN at S398 and S113 (Bassi et al., 2013, Chen et al., 2015). In both cases, the phosphorylation affects the nuclear localization of PTEN. Although it cannot be ruled out that ATM affects PTEN activity, no indication of such has been published to date. We never analysed if activation of ATM by neuronal activity and/or stress affects*

PTEN localization or activity. In any case, if ATM's function during neuronal activity would induce both responses - translocation of PTEN into the nucleus AND increasing phosphorylation of pS647-DBN – we would expect the same outcome. We believe it would be a very interesting idea to pursue in future studies.

- We have tried with different methods to specify different pools of DBN using a number of techniques (including FUNCAT-PLA). However, at this stage, we find it difficult to precisely locate the newly synthesized DBN pool that may function as reserve for stabilization.*

Line 59, “cellular insult or stress renders spines vulnerable to DBN-loss”, surely the other way around?

- We believe the sentence would work also the other way around – but have now changed the text accordingly.*

Line 122, HEK 293T

- We thank the reviewer for spotting this wrong statement, it has been modified accordingly*

Line 160, absence and or presence

- Has been changed to “in the absence or presence”*

Line 247, why is Fig. S5 before Fig. S4? (line 254)

- This has been changed.*

Line 309, 30 ng

- Has been changed accordingly*

Line 471, were PTEN antibodies used?

- No PTEN antibodies were used and the name of the antibody has been removed from the section*

Line 754, n.s. not used in figure

- This has been changed accordingly*

Figure 5a, why are there two bands for drebrin in the immunoblot and which one is hDBN-YFP? Probably the 130 kDa band as judged by the blot in Fig S3 (N2), although there is also a band at 100 kDa in this blot. The 180 kDa band can't be dbn-1 because this is a □dbn-1!!

- This issue has also been raised by Reviewer 2. The additional band in the previous manuscript resulted from proteolysis. We have optimized the lysis protocol to avoid any processing of the human DBN-YFP. We have now included further protease inhibitors (see Materials and Methods) to avoid any degradation during lysis and processing of the samples prior to SDS-PAGE and western blot analysis. Figure S3g depicts the human DBN variants expressed in the nematode and only a single protein band is now visible for $nDBN^{wt}$, $nDBN^{S647A}$ and $nDBN^{S647D}$.*

Reviewer #4 (Remarks to the Author):

1. In Fig. 2C's pulse chase experiments, authors used HEK293T cells and transfected DBNs to determine the effect of s65 phosphorylation on its half-life. In Fig. 4, the effect of oxidation-induced ATM activation for DBN phosphorylation was examined in HEK293T cells that express ATM and DBN endogenously (Page 7, line 167). It was clear (from methods or the text) whether there are two different HEK293T cell lines that exhibit difference in the endogenous level of DBN and ATM, or they are really just the same line, and authors examined the endogenous DBN protein in the second experiment. If this is the case, it may be more relevant to determine the half-life of protein at the endogenous level, although it does make comparison of the half-life of DBN S65D or DBN S65A difficult.

- *We thank the reviewer for pointing out this inconsistency. The cells used in fig. 2c, fig. 2d and in fig. 4c were all the same HEK293T cells. In fig. 2c and fig. 2d we transfected FLAG-DBN mutants to study the influence of S647 phosphorylation on DBN half-life. In fig.4C, we studied endogenous DBN. For clarity, we have now inserted the cell line used in the legend of fig.2d, which was missing before.*

2. Using ATM^{-/-} mice, as well as the ATM inhibitor, authors convincingly demonstrated a requirement of ATM for stress- or activity-dependent increase of S65 DBN phosphorylation. Is there any expression data that corroborate the either steady-state or activity-dependent localization of ATM to dendrites? In addition, steady-state phosphorylation at s65-DBN seems to be dependent on other, unidentified kinases. Authors should make this clear to readers.

- *This criticism has also been raised by reviewer 1. Please compare with point 8 concerning the localization of ATM in neurons. We have included a number of material and discuss this issue at length there (page 5 of this rebuttal letter).*

3. Comparatively, the *C. elegans* studies are not solid or as well presented, and it needs a bit more work.

- *In the revised manuscript we have extended figure 6 (figure 5 in previous version of the manuscript) and Supplemental figure S3 and S4 to increase the systematic and extent in which we analyzed *C. elegans dbn-1* partial and complete mutants, as well as the transgenic nematode lines expressing human DBN and its (de)-phospho-mimicry mutants.*

I am impressed that in addition to order a deletion strain from the stock center that does not lead to a complete KO (ok925), authors generated a true knockout of the *C. elegans* homologue by CPRISPR (JKM1) in order to have true complete KO. But authors never presented data to compare the phenotype of the two strains, or the hinted successful removal of residual function of DBN by feeding RNAi in the ok925 background (Methods; Page 21). This is critical because authors used different genetic background to generate the 'humanized' DBN strains. They expressed DBNS65A or D proteins in ok925, whereas DBN wild-type in the JKM1 background (Methods; Page 1). To compare their effect on aging and oxidative stress fairly, authors should make sure that the background (*C. elegans* DBN^{-/-}) should be equal, and the expression level of the three integrated transgenes to be at least similar. There should be quantified data for this.

- *We thank the reviewer for his/her insightful criticism and comments. Firstly, he/she would like to obtain further characterization of the *dbn-1* mutant strains. We therefore undertook further characterization of RB1004 (QL100; partial *dbn-1* mutant expressing a residual DBN-1 fragment), RB1004 combined with *dbn-1* RNAi to demonstrate the complete depletion of *dbn-1* and JKM1 (CRISPR generated complete knockout of *dbn-1*). We have now added a western blot analysis of the nematode DBN-1 protein levels in these different mutant and knockdown strains. The western blots are depicted in supplemental figures S3 and S4.*

- *Secondly, the reviewer requested an analysis of the expression levels of the human DBN variants (DBN^{wt}, DBN^{S647A} and DBN^{S647D}) in the RB1004 background. We have now analyzed the expression levels by fluorescence microscopy and western blot of the YFP tagged humanized DBN nematodes and show the data in supplemental figure S3.*

Authors did not show whether the elegans dbn mutants have an aging-related phenotype when compared to wild-type animals. This is critical. The conclusion of these studies should suggest that dbn mutants exhibited reduced aging, or more sensitivity to PA-induced aging. Overall, this part of the study reads like an addition that ignores the rich biology of an organism, and reduces it simply to be a test-tube to compare the effect of overexpressing vertebrate DBNs by a convenient assay. Though this may be a trend of thinking of the neurodegenerative field, it is still a bit painfully wasteful way to use this system for aging, because it has so much more to offer for true mechanistic discoveries when used properly.

- *We agree with the reviewer and thank him/her for this critical comment. The reviewer suggests a more thorough analysis of the C. elegans dbn-1 mutants with respect to aging. In response, we performed an extensive analysis of lifespan assays of RB1004 (partial knockout of dbn-1), RB1004 + RNAi of dbn-1, N2 and JKM1 (CRISPR generated complete knockout of dbn-1). We analyzed the lifespan of the same strains upon treatment with paraquat. The data are depicted in supplemental figure S3a-e and we can demonstrate that the dbn-1 mutant exhibits indeed a paraquat-induced aging phenotype. We can also demonstrate that the partial knockout strain (RB1004) in combination with RNAi of dbn-1 shows the same paraquat-induced aging phenotype as the CRISPR generated complete dbn-1 knockout strain JKM-1 and therefore justifies using the RB1004 + RNAi of dbn-1 as genetic background for the expression of DBN^{wt}, DBN^{S647A} or DBN^{S647D} (figure S3d+e).*

In addition, we analyzed the produced viable offspring as readout for organismal fitness of the wild type N2, RB1004 (partial knockout of dbn-1), N2 + RNAi of dbn-1 and the complete knockout of dbn-1 (JKM1). The depletion of dbn-1 by either RNAi or CRISPR leads to a significant reduction of about 20% compared to the wild type or partial knockout of dbn-1 (figure S3f).

Similar to the above, the dependence of the DBN phenotypes on ATM was examined by the ATM inhibitor, which was only partially effective. Caveats remain regarding its specificity or efficacy to the endogenous C. elegans ATM. Authors could at least discuss this caveats, and better approaches, such as using atm ko mutants, for future studies if they do want to address this process using the C. elegans experimental system.

- *The reviewer points out that a lack of specificity and/or efficacy of the ATM inhibitor could be a potential caveat for the analysis in C. elegans. To address this concern, we have validated the efficacy of the ATM inhibitor and show in figure S4c a significant decrease of the phosphorylation of nDBN using the pS647-DBN antibody as readout (quantification now included). In addition, we observed a reduction in the median half-live of the WT nDBN to the same median lifespan of the dephospho-DBN^{S647A} variant (figure 6c-e) and thus feel confident that the ATM inhibitor KU55933 is active and efficient for the C. elegans system. We cannot exclude potential side effects due to a lack of specificity. However, we did not observe any obvious differences to controls. In fact, KU55933 treatment has no effect on the lifespan in the absence of paraquat and thus arguing against any unspecific effects (figure 6c-e).*

4. Overall the manuscript is prepared in such compact format, while very well prepared, had little room to discuss backgrounds and caveats, and so much experimental details that are critical for the interpretation of the results, such as the specific cell culture type, their different DIV stage, different genetic backgrounds, were all left at the Methods for readers to find. Authors should reformat the manuscript for Nature Communications so that proper results sections can be read to help both the readers understand and appreciate the rigor that went into each set of their experiments.

- *We have thoroughly revised the manuscript and included more data as requested by all reviewers, in particular in the C. elegans section. We have also revised the data presentation and think that the manuscript has gained in clarity thanks to the reviewer's comments.*

Just as another example to illustrate this point: In both Fig. 5 and Fig. S5, DBN antibodies revealed two bands, 180Kd and 130Kd, respectively. Are they two isoforms of DBN that both could be phosphorylated at S65? It seems so but there was no description or explanation of such information that I could easily find. If this is the case though, it was not explained in all other western blot, which only the 130kd isoform was examined in all biochemistry experiments shown in the paper.

- *This concern is shared by reviewers 2 and 3. The additional band in the previous manuscript resulted from proteolysis. We have optimized the lysis protocol to avoid any processing of the human DBN-YFP. We have now included a cocktail of protease inhibitors (see Materials and Methods) to avoid any degradation during lysis and processing of the samples prior to SDS-PAGE and western blot analysis. Figure S3g depicts the human DBN variants expressed in the nematode and only a single protein band is now visible for $nDBN^{wt}$, $nDBN^{S647A}$ and $nDBNS^{647D}$.*

REVIEWERS' COMMENTS:

Reviewer #1 (Remarks to the Author):

The authors have addressed the comments in my initial review satisfactorily. There are still some grammatical errors throughout. One that came up repeatedly was the use of "half-live" incorrectly for the singular of "half-life".

Reviewer #2 (Remarks to the Author):

This study by Kreis et al was reviewed very positively by me during its first submission, and I still believe that the study would be -in principle- highly suitable for publication in Nature Communications. However, disappointingly, the revision/response addressed my two major concerns rather sloppily: As indicated below, major concern 1 remains in full; while major concern 2 has been addressed for the most part, some additional minor changes are still required.

1)I remain concerned regarding the inclusion of "lifespan" in the title, especially without qualifier. This is because the data do NOT show general extension of lifespan, but instead a protection from continued oxidative stress.

The authors responded by stating "We believe that the inclusion of 'lifespan' in the title is justified, as it is a standard readout using in the scientific community of researchers using the C. elegans model. We believe that by demonstrating an effect of DBN phosphorylation during oxidation induced stress, our results can be referred to an experimental setting that tests life-span".

This response does not adequately address my concern. For instance, lifespan is also commonly used as a readout in cancer research, yet the unqualified statement that a specific cancer treatment "increases lifespan" in general (i.e. without adding "in xyz cancer") would be considered false in the scientific community of cancer researchers. I doubt that this is different in the community of researchers using the C. elegans model; while I may be wrong about this, the authors do not provide any citations to dissipate this doubt. And even if I am wrong, the qualifier "in C. elegans" should be added. (That is, if it is indeed common practice to add oxidative stress in any lifespan study in C. elegans, without qualifying the results as being protective from oxidative stress and instead pertaining to lifespan in general).

2)In the response to major point 2, I do not understand why the authors provided the citation that they picked: This paper shows effects on spines by half the concentration used in the current manuscript (thus underscoring my point, i.e. that this concentration should have an effect, and that I am not aware

of any publication that showed no effect on spines at such high concentrations). To the changed text on page 3 in response to major point 2: “On the contrary” does not make syntactical sense. More importantly, I would strongly suggest to state “we did not observe... in our preparation” instead of “we do not observe...”. This is because pretty much everyone sees spine shrinkage at 1 μ M Abeta (although some test after times longer than 24 h). Also, I would suggest revisiting the choice of citation for the statement that 1 μ M (or less) typically induces spine loss (perhaps citing at least two papers and including work from the pioneers of the field).

Minor points:

- 1) Figure 1a: CaMKII misspelled in the Figure panel; in the legend, the changed (highlighted yellow sentence) doesn't make sense (i.e. spine detection by colocalization of something with something else that is located in spines as well as in dendrites).
- 2) Figure 1 e legend: typo in highlighted added text. “1 μ M”, not “1uM”, consistent with other use. Same with insertion of “1uM” on page 3.
- 3) Figure 1e legend still includes “globular” in the description of the Abeta oligomers (in contrast to the author's statemnt in response to previous minor point 10).

Reviewer #3 (Remarks to the Author):

In their revised manuscript Kreis et al., have responded fully and adequately to all of my reviewer's comments and I see no reason why the manuscript should not now be accepted for publication.

Reviewer #4 (Remarks to the Author):

The authors have satisfactorily addressed my questions and comments.

- 1) They have expanded the *C. elegans* experiments and clarified the phenotypic ambiguity of the genetic backgrounds of different *dbn-1* mutants. The phenotype of the true *dbn-1* knockout mutants suggests a physiological, protective role of DBN-1 under stress (Fig. S3). The outcome of comparing the effect of different humanized worms expressing different forms of DBNs was not as clean, though as expected –

not only due to the differences in the expression level and the differences in the host genomic background, but also, the potential messiness of overexpressing phosphor-'dead' and phosphor-mimic forms of a protein in vivo. To me, however, this is not as critical as demonstrating a clean, specific, and physiological phenotype of dbn-1 knockout mutants upon paraquat-treatment.

I hope that the authors will consider the following suggestions:

a- I strongly recommend to rearrange Figure 6 and Figure S3. Fig. 6a to me is a supplementary figure panel (which goes nicely with Fig. 3sg). The more important key data, which were shown in Figure S3a-f, was that the loss of DBN-1 does not cause an aging phenotype under normal conditions, but leads to less tolerance to paraquat treatment. This information can be compressed and moved to the main figure to replace Figure 6a.

b- Fig. S3d does not include the lifespan of untreated RB1004; dbn-1 animals. Is this intentionally omitted or just an error? It would be good to include data from this experimental group in the figure.

c- Fig. S3e, the lifespan of untreated - RB1004 kd;dbn-1 (13.7) vs N2 kd::dbn-1 (12.7) does not seem to be consistent with what was shown in the graph panel (Fig. S3b)

d- Fig. 6e: authors should clearly specify in this panel that two different delta dbn-1 backgrounds were used to construct the 3 different humanized models. Current panel labeling can easily mislead that all transgenic animals were in the same background. To my understanding, RB1004 was used to express the two mutated versions of DBN, and JKM14 was used to express the WT form of DBN. This information should be clarified in both the figure panel and legend.

2) They cleaned up the biochemistry experiments (eliminated the proteolytic product) of some western blot analyses shown in the previous version. Well done.

The current manuscript is further strengthened, and I fully support its acceptance for publication upon addressing comments and suggestions in 1). Thank you for the nice work.

RESPONSE TO REVIEWERS' COMMENTS:

Reviewer #1

The authors have addressed the comments in my initial review satisfactorily. There are still some grammatical errors throughout. One that came up repeatedly was the use of "half-live" incorrectly for the singular of "half-life".

We thank the reviewer for raising this issue, and especially spotting the misuse of 'half-live'. We changed 'half-live' to 'half-life' throughout the document and undertook a careful spelling check.

Reviewer #2

1) I remain concerned regarding the inclusion of "lifespan" in the title, especially without qualifier. This is because the data do NOT show general extension of lifespan, but instead a protection from continued oxidative stress.

After some careful consideration, we agree with the reviewer's concern and have changed the title to 'ATM phosphorylation of the actin-binding protein drebrin controls oxidation stress-resistance in mammalian neurons and C. elegans'. We hope that the reviewer find this new title suitable.

2) In the response to major point 2, I do not understand why the authors provided the citation that they picked: This paper shows effects on spines by half the concentration used in the current manuscript (thus underscoring my point, i.e. that this concentration should have an effect, and that I am not aware of any publication that showed no effect on spines at such high concentrations). To the changed text on page 3 in response to major point 2: "On the contrary" does not make syntactical sense. More importantly, I would strongly suggest to state "we did not observe... in our preparation" instead of "we do not observe...". This is because pretty much everyone sees spine shrinkage at 1 μ M A β (although some test after times longer than 24 h). Also, I would suggest revisiting the choice of citation for the statement that 1 μ M (or less) typically induces spine loss (perhaps citing at least two papers and including work from the pioneers of the field).

We have modified the previous section

From:

'To analyse if loss of DBN increases susceptibility to the synaptotoxic effects of A β ₁₋₄₂, we challenged DBN^{-/-} neurons with 1 μ M amyloid peptide. Although typically 1 μ M amyloid peptide has been shown to decrease dendritic spine density of hippocampal rat neurons¹², we do not observe any changes in spine density of WT mouse neurons at this concentration. On the contrary, we saw a significant reduction in spine density in DBN^{-/-} neurons (Fig. 1e).'

To:

'To analyse if loss of DBN increases susceptibility to the synaptotoxic effects of A β ₁₋₄₂, we challenged hippocampal neurons with amyloid peptide oligomeric preparations¹². At concentrations that had previously been demonstrated to induce spine loss in rat hippocampal neurons (1 μ M or lower)^{13,14}, the amyloid peptide did not induce significant decreases in spine density in our mouse hippocampal neurons. We believe this discrepancy is likely due to species differences (rat versus mouse), or normal variations in neuron preparations or peptide oligomerisation. However, we found a significant reduction in spine density in Dbn^{-/-} neurons with 1 μ M amyloid peptide (Fig. 1e), indicating that drebrin loss renders spines more vulnerable towards synaptotoxic effects of A β ₁₋₄₂.

Minor points:

1)Figure 1a: CaMKII misspelled in the Figure panel; in the legend, the changed (highlighted yellow sentence) doesn't make sense (i.e. spine detection by colocalization of something with something else that is located in spines as well as in dendrites).

We have now spelt CaMKII in the correct way in figure 1A as well as in the method section.

We appreciate the reviewer's notion that CaMKII cannot be used as a specific spine marker. It seems we didn't convince the reviewer that we used CaMKII merely as an additional marker protein to detect populations of larger dendritic spines. Whilst CaMKII accounts for 2–6% of total protein in PSDs only, it exceeds the relative protein level of the prototypal postsynaptic scaffold protein PSD-95 especially in larger PSDs (Hell, Neuron 2014). Therefore, we found it informative to include that F-actin-rich spines detected by our analysis software also display intense CaMKII labelling, indicating that we predominately detected PSD-comprising spines and not filopodia.

2)Figure 1 e legend: typo in highlighted added text. "1 uM", not "1uM", consistent with other use. Same with insertion of "1uM" on page 3.

We have corrected this mistake in Fig 1e legend as well as in the text page 3.

3)Figure 1e legend still includes "globular" in the description of the Abeta oligomers (in contrast to the author's statement in response to previous minor point 10).

We apologize for this omission. We had changed globular to synaptotoxic A β species in the material and method section, but forgot to change it in the figure legend. We have now changed globular to oligomeric A β ₁₋₄₂ in both the material and method section and in the figure legend.

Reviewer #3

In their revised manuscript Kreis et al., have responded fully and adequately to all of my reviewer's comments and I see no reason why the manuscript should not now be accepted for publication.

We thank the reviewer for this positive response

Reviewer #4

The authors have satisfactorily addressed my questions and comments.

1) They have expanded the C. elegans experiments and clarified the phenotypic ambiguity of the genetic backgrounds of different dbn-1 mutants. The phenotype of the true dbn-1 knockout mutants suggests a physiological, protective role of DBN-1 under stress (Fig. S3). The outcome of comparing the effect of different humanized worms expressing different forms of DBNs was not as clean, though as expected – not only due to the differences in the expression level and the differences in the host genomic background, but also, the potential messiness of overexpressing phosphor-'dead' and phosphor-mimic forms of a protein in vivo. To me, however, this is not as critical as demonstrating a clean, specific, and physiological phenotype of dbn-1 knockout mutants upon paraquat-treatment.

I hope that the authors will consider the following suggestions:

a- I strongly recommend to rearrange Figure 6 and Figure S3. Fig. 6a to me is a supplementary figure panel (which goes nicely with Fig. 3sg). The more important key data, which were shown in Figure S3a-f, was that the loss of DBN-1 does not cause an aging phenotype under normal conditions, but leads to less tolerance to paraquat treatment. This information can be compressed and moved to the main figure to replace Figure 6a.

We have rearranged the figures as suggested:

Revised Figure 7 (previous Figure 6):

- a - Lifespan of N2 vs RB1004 vs JKMI (\pm Paraquat), former S3d
- b - Progeny assay, former S3f
- c – e - former Fig.6 b – d

The table showing half-lives (former S3e) has been moved to a separate Table (Table 1) (as required by the Editorial Requests). The same applies to the former table in figure 6e which is also now a separate table named Table 2.

Revised Supplemental Figure 3:

- a - Confocal images of human Drebrin *C. elegans* lines, former Fig.6a
- b - Western blot nDBN-Quantification + Graph, former S3g
- c - Fluorescence nDBN-Quantification, former S3h
- d - Lifespan N2 vs RB1004 (\pm Paraquat), former S3a
- e - Lifespan N2 vs RB1004 (\pm knockdown), former S3b
- f - Lifespan N2 vs RB1004 (+ Paraquat, \pm knockdown), former S3c

b- Fig. S3d does not include the lifespan of untreated RB1004; dbn-1 animals. Is this intentionally omitted or just an error? It would be good to include data from this experimental group in the figure.

We apologize for this error. We now have included the lifespan of untreated RB1004 kd:dbn-1 in the figure (now Figure 7a).

c- Fig. S3e, the lifespan of untreated - RB1004 kd;dbn-1 (13.7) vs N2 kd::dbn-1 (12.7) does not seem to be consistent with what was shown in the graph panel (Fig. S3b)

The value for the half-life of RB1004 dbn-1 knockdown is 10.7 days. We have corrected the table displayed the data now in Table 1 accordingly.

d- Fig. 6e: authors should clearly specify in this panel that two different delta dbn-1 backgrounds were used to construct the 3 different humanized models. Current panel labeling can easily mislead that all transgenic animals were in the same background. To my understanding, RB1004 was used to express the two mutated versions of DBN, and JKMI14 was used to express the WT form of DBN. This information should be clarified in both the figure panel and legend.

Indeed, different genetic backgrounds were used. Total dbn-1 knockout (JKMI) for wt human Drebrin and partial knockout (RB1004) + dbn-1 RNA-mediated knockdown for phospho/dephospho-mutants of human Drebrin.

For the sake of clarity and simplicity we referred to both as Drebrin depletion. Nevertheless, we acknowledge the point that this could be misleading and changed the presentation accordingly in the following table (header of Table 2):

	+nDBN variants				
	no DBN	RB1004 kd:dbn-1	wt JKMI kd:dbn-1	S647A RB1004 kd:dbn-1	S647D RB1004 kd:dbn-1
non treated	13.3		13.3	13	14
+ KU55933	13.7		13.7	13.1	15.5
+ paraquat	8.3		11.8	9.7	11.8
+ paraquat + KU55933	11		9.7	9.6	11.7

2) They cleaned up the biochemistry experiments (eliminated the proteolytic product) of some western

blot analyses shown in the previous version. Well done.

The current manuscript is further strengthened, and I fully support its acceptance for publication upon addressing comments and suggestions in 1). Thank you for the nice work.